



# Field measurements of trace gases and aerosols emitted by peat fires in Central Kalimantan, Indonesia during the 2015 El Niño

Chelsea E. Stockwell[1,7], Thilina Jayarathne[2], Mark A. Cochrane[3], Kevin C. Ryan[4,8], Erianto I. Putra[3,5], Bambang H. Saharjo[5], Ati D. Nurhayati[5], Israr Albar[5], Donald R. Blake[6], Isobel J. Simpson[6], Elizabeth A. Stone[2], Robert J. Yokelson[1]

[1]University of Montana, Department of Chemistry, Missoula, 59812, USA

[2]University of Iowa, Department of Chemistry, Iowa City, 52242, USA

[3]South Dakota State University, Geospatial Sciences Center of Excellence, Brookings, 57006, USA

[4]United States Forest Service, Missoula Fire Sciences Laboratory (retired), Missoula, 59808, USA

[5]Bogor Agricultural University, Faculty of Forestry, Bogor 16680, ID

[6]University of California, Irvine, Department of Chemistry, Irvine, 92697, USA

[7]Now at: Chemical Sciences Division, NOAA Earth System Research Laboratory, Boulder, 80305, USA

[8]Now at: FireTree Wildland Fire Sciences, L.L.C., Missoula, 59801, USA

*Correspondence to*: R. J. Yokelson (bob.yokelson@umontana.edu)

## Abstract

Peat fires in Southeast Asia have become a major annual source of trace gases and particles to the regional-global atmosphere. The assessment of their influence on atmospheric chemistry, climate, air quality, and health has been uncertain partly due to a lack of field measurements of the smoke characteristics. During the strong 2015 El Niño event we deployed a mobile smoke sampling team in the Indonesian province of Central Kalimantan on the island of Borneo and made the first, or rare, field measurements of trace gases, aerosol optical properties, and aerosol mass emissions for authentic peat fires burning at various depths in different peat types. This paper reports the trace gas and aerosol measurements obtained by Fourier transform infrared spectroscopy, whole air sampling, photoacoustic extinctiometers (405 and 870 nm), and a small subset of the data from analyses of particulate filters. The trace gas measurements provide emission factors (EFs, g compound per kg biomass burned) for $CO_2$, CO, $CH_4$, non-methane hydrocarbons up to $C_{10}$, 15 oxygenated organic compounds, $NH_3$, HCN, $NO_x$, OCS, HCl, etc.; up to ~90 gases in all. The modified combustion efficiency (MCE) of the smoke sources ranged from 0.693 to 0.835 with an average of 0.772 ± 0.053 (*n*=35) indicating essentially pure smoldering combustion and the emissions were not initially strongly lofted. The major trace gas emissions by mass (EF as g/kg) were: carbon dioxide (1564 ± 77), carbon monoxide (291 ± 49), methane (9.51 ± 4.74), hydrogen cyanide (5.75 ± 1.60), acetic acid (3.89 ± 1.65), ammonia (2.86 ± 1.00), methanol (2.14 ± 1.22), ethane (1.52 ± 0.66), dihydrogen (1.22 ± 1.01), propylene (1.07 ± 0.53),





propane (0.989 ± 0.644), ethylene (0.961 ± 0.528), benzene (0.954 ± 0.394), formaldehyde (0.867 ± 0.479), hydroxyacetone (0.860 ± 0.433), furan (0.772 ± 0.035), acetaldehyde (0.697 ± 0.460), and acetone (0.691 ± 0.356). These field data support significant revision of the EFs for $CO_2$ (–8%), $CH_4$ (–55%), $NH_3$ (–86%), CO (+39%) and other gases compared with widely-used recommendations for tropical peat fires based on a lab study of a single

sample published in 2003. BTEX compounds (benzene, toluene, ethylbenzene, xylenes) are important air toxics and aerosol precursors and were emitted in total at 1.5 ± 0.6 g/kg. Formaldehyde is probably the air toxic gas most likely to cause local exposures that exceed recommended levels. The field results from Kalimantan were in reasonable agreement with recent (2012) lab measurements of smoldering Kalimantan peat for "overlap species," lending importance to the lab finding that burning peat produces large emissions of acetamide, acrolein, methylglyoxal, etc.,

which were not measureable in the field with the deployed equipment and implying value in continued similar efforts.

The aerosol optical data measured include EFs for the scattering and absorption coefficients (EF $B_{scat}$ and EF $B_{abs}$, $m^2$/kg fuel burned) and the single scattering albedo (SSA) at 870 and 405 nm, as well as the absorption Ångström exponents (AAE). By coupling the absorption and co-located trace gas and filter data we estimated black carbon

(BC) EFs (g/kg) and the mass absorption coefficient (MAC, $m^2$/g) for the bulk organic carbon (OC) due to brown carbon (BrC). Consistent with the minimal flaming, the emissions of BC were negligible (0.0055 ± 0.0016 g/kg). Aerosol absorption at 405 nm was ~52 times larger than at 870 nm and BrC contributed ~96% of the absorption at 405 nm. Average AAE was 4.97 ± 0.65 (range, 4.29-6.23). The average SSA at 405 nm (0.974 ± 0.016) was marginally lower than the average SSA at 870 nm (0.998 ± 0.001). These data facilitate modeling climate-relevant

aerosol optical properties across much of the UV/visible spectrum and the high AAE and lower SSA at 405 nm demonstrate the dominance of absorption by the organic aerosol. Comparing the $B_{abs}$ at 405 nm to the simultaneously measured OC mass on filters suggests a low MAC (~0.1) for the bulk OC, as expected for the low BC/OC ratio in the aerosol. The importance of pyrolysis (at lower MCE), as opposed to glowing (at higher MCE), in producing BrC is seen in the increase of AAE with lower MCE ($r^2$ = 0.65).

## 1 Introduction

Many major atmospheric sources have been studied extensively with a wide range of instrumentation. This includes, for example, temperate forest biogenic emissions (e.g. Ortega et al., 2014) and developed-world fossil-fuel based emissions (e.g. Ryerson et al., 2013). Biomass burning (BB) is the second largest global emitter of $CO_2$, total greenhouse gases, and non-methane organic gases (NMOGs), with the latter being precursors for ozone ($O_3$) and

secondary organic aerosol (OA). BB is the largest global source of fine primary OA, black carbon (BC) and brown carbon (BrC) (Akagi et al., 2011; Bond et al., 2004, 2013). However, many important, complex, BB emission sources have been rarely, if ever, characterized by comprehensive field measurements (Akagi et al., 2011). The largest of these undersampled BB sources is peatland fires, which occur primarily in boreal forests and in the tropics, especially the Indonesian provinces of Sumatra, Kalimantan, and Papua as well as Malaysian Borneo.

Peatland fires in the tropics usually start in surface fuels with surface fuel consumption commonly ranging from ~1-20 MgC/ha as a result of land-clearing and agricultural activities common throughout the tropics (Page et al., 2009;





Akagi et al., 2011). As the surface fuels are consumed, the much larger store of belowground biomass (mostly peat) at loadings of ~500-600 MgC/ha per meter depth, and up to 20 m deep, can become ignited and propagate as a glowing front that dries and pyrolyzes the fuel ahead of it (Yokelson et al., 1997; Page et al., 2002; Usup et al., 2004; Huang et al., 2016). Once the glowing fronts are burning under a layer of ash or have undercut the peat, the

fire is virtually impossible to extinguish by commonly available means and it can burn slowly, both horizontally and downward to the water table for months. Peat fires can also re-emerge and ignite surface fuels, but the smoldering consumption of large quantities of belowground fuel, which produces smoke that is initially weakly-lofted, is a key ecological and atmospheric characteristic of peatland fires (Tosca et al., 2011).

The local air quality impacts of peat fires can be dramatic. As an example $PM_{10}$ levels in Palangkaraya, Indonesia

reached 3741 ug/m$^3$ on 20 October, 2015 (BMKG, 2015) during a months-long pollution crisis that had simultaneous counterparts in Sumatra and Papua. With unfavorable transport, locally-generated smoke may be dispersed to numerous major population centers regionally where much reduced but more widespread exposure and health effects are a potential concern (e.g. Aouizerats et al., 2015).

Since peat is a semi-fossilized fuel (accumulation rates are a few mm per year; Wieder et al., 1994; Page et al.,

1999), the impacts on the carbon cycle are larger for the same amount of biomass burned than for most other BB types, and the carbon emissions may be significant in comparison to total fossil fuel carbon emissions in some years (e.g. 13-40% in 1997, Page et al., 2002). In Southeast Asia, in the 1980s and 1990s, peatland fires were a major source of carbon to the atmosphere mainly during El-Niño-induced droughts when fire danger was higher, the fire season was longer, and water tables were lower. With accelerated deforestation and building of drainage canals (e.g.

4000 km of canals as part of the Ex Mega Rice Project (EMRP) started in 1996 (Putra et al., 2008; Hamada et al., 2013)), peat fires and their impacts are now extensive on an annual basis (van der Werf et al., 2010; Wiedinmyer et al., 2011; Gaveau et al., 2014). In many disturbed areas the absence of the original peat swamp forest's moist under-canopy microclimate that acted to deter ignition or slow fire spread results in increased fire activity (Cochrane et al., 1999). In these areas ferns, plantations, or patches of secondary forest overlie peat that has often already been

impacted by previous fires and/or by roads and canals that also increase access and fire activity. The disturbed-area surface fuels are usually a minor component of the total available fuel, but are present in sufficient amounts to be an ignition source for the peat.

Previously, tropical peat fire emissions had only been measured in detail in a few laboratory experiments (e.g. Christian et al., 2003) and most recently during the fourth Fire Lab at Missoula Experiment (FLAME-4, Hatch et al.,

2015; Jayarathne et al., 2014; Stockwell et al., 2014, 2015). The lab emissions measurements featured an extensive suite of instruments, many of which would be difficult to deploy in remote field conditions, but the realism of the lab burning conditions was hard to judge except qualitatively/visually. Further, the emissions from burning one peat sample from Sumatra (Christian et al., 2003) were quite different from the average emissions generated by burning three samples of Kalimantan peat during FLAME-4. For example, the "Sumatra/Kalimantan" emission ratio was ~2

for $CH_4$ and ~11 for $NH_3$ (Stockwell et al., 2014). This variability makes it unclear how to optimize regional emissions inventories and the mean and variability in lab studies could also potentially reflect artifacts arising from sample collection, storage, or handling procedures. As a result, field measurements were a critical priority.





Beginning in 2013, an international team involving South Dakota State University, Bogor Agricultural University (IPB), the University of Montana, University of Iowa, University of California at Irvine, the United States Forest Service, and the Borneo Orangutan Survival Foundation (a Kalimantan NGO) initiated a multi-faceted study of peat fires in the Central Kalimantan province of Indonesia. The activities built on earlier work by the Kalimantan Forest

and Climate Partnership (KFCP, Applegate et al., 2012; Ichsan et al., 2013, Graham et al., 2014a, b; Hooijer et al., 2014) established in 2009 and included fire-scene investigations; fire history documentation; vegetation and fuels mapping; hydraulic conductivity, water table, and subsidence monitoring with an extensive series of 515 wells and 81 subsidence poles along 70 km of transects; collecting peat samples for the FLAME-4 laboratory emissions measurements; burned area mapping; and Lidar transects to quantify depth of burn (Ballhorn et al., 2009). In this

paper we present our October-November, 2015 ground-based field measurements of trace gases and aerosols directly in 35 different peat fire plumes in the vicinity of Palangkaraya, Central Kalimantan, in the mostly-disturbed western part of the EMRP (Page et al., 2002, 2009; Usup et al., 2004). We describe the sampling sites, peat characteristics, and our instrument selection, which aimed to optimize the trade-offs between the required mobility and the need for detailed measurements to understand atmospheric impacts and compare with a suite of "overlap

species" also measured in the FLAME-4 lab studies. We present and discuss our trace gas emission factors (EFs, g compound produced per kg peat burned) measured by a cart-based, mobile Fourier transform infrared spectrometer (FTIR) and by filling whole air sampling (WAS) canisters for subsequent lab analyses. The EFs provided include: $CO_2$, $CO$, $NH_3$, $NO_x$, $CH_4$, and numerous non-methane organic gases (NMOG) up to $C_{10}$, as many as ~90 gases in all. We present and discuss our measurements of aerosol optical properties and mass measured by photoacoustic

extinctiometers (PAX) and gravimetric filter sampling. The aerosol data include: EFs for scattering and absorption coefficients (EF $B_{scat}$, EF $B_{abs}$, $m^2$/kg peat burned) at 870 and 405 nm, the single scattering albedo (SSA) at 870 and 405 nm, and the absorption Ångström exponents (AAE). These data facilitate modeling of aerosol optical properties across much of the UV-visible spectrum. We also present and discuss BC emission factors (g/kg fuel burned) and the mass absorption coefficient (MAC, $m^2$/g) for the bulk organic carbon (OC) due to BrC emissions that are based

on combining the PAX absorption data with co-located trace gas and filter measurements. Our field measurements enable us to assess emissions of the main greenhouse gases emitted by fires, many ozone and organic aerosol precursors, several air toxics, and the absorbing BrC that dominates the direct radiative forcing of peat fire smoke. Finally, we compare our field data to lab results published in 2003, IPCC guidelines, and the recent FLAME-4 lab measurements of burning Indonesian peat to gain additional insight into the emissions of air toxics and precursors

not measured in the field and assess the overall value of lab studies of burning peat. Additional aerosol results based on our filter sampling in the field coupled with a large suite of subsequent analyses will be reported in a companion paper (Jayarathne et al., 2016, in preparation).



## 2 Experimental details

### 2.1 Site descriptions

Peat is an accumulation of partially decayed vegetation or organic matter that can be further classified as fibric, hemic, or sapric (by increasing degree of decomposition and density (Wüst et al., 2003)). Different amounts of roots; sound or rotten logs; charred logs, char, and ash from previous burns; and mineral soil are frequently mixed in with the peat along with varying amounts of water. On undisturbed sites deeper peat is normally more decomposed and denser, but on disturbed sites the upper layer is sometimes already removed by previous fires, while dredging for canals can place "older peat" on top of younger peat, and road-building can compact the peat. Traditional peat classification schemes can be less straightforward for disturbed areas. For instance, ferns and grasses can contribute fibrous roots to a layer of older, even sapric, material. We note that the Kalimantan peat burned in the FLAME-4 lab study that we will compare to was sampled in both undisturbed forest ($n$=1) and previously logged/burned forest ($n$=1), whereas the peat fires sampled in this field work were all on moderately to heavily disturbed sites, which is generally where fire activity is the highest.

Peat can burn at above 100% fuel moisture (defined as $100 \times$ (wet–dry)/dry) because the glowing front pre-dries the fuel as it advances. Peat combustion can occur as a glowing front in an expanding pit or undercut, but with direct access to surface air (Huang et al., 2016), which we term "lateral spreading." The glowing front can be covered by ash or initially propagate downward on inclusions or in cracks in initially, mostly-unburned peat, which we refer to as "downward" spreading, but this is much less common. Figure S1 shows photographs of these spread modes. The glowing front is the site of gasification reactions ($O_2$-oxidation of char) that produce mostly $CO_2$, CO, $CH_4$, $NH_3$, and little visible aerosol. The heat from glowing combustion pyrolyzes the adjacent peat, producing relatively more organic gases and copious amounts of white smoke (with high OA content) (e.g. Fig. 3 in Yokelson et al., 1997). Wind increases the glowing front temperature. Oxygen availability is likely higher for lateral spreading than downward spreading fire and the overburden in downward spreading fires may scavenge some emissions. Occasionally peat can support brief, small flames if the surface peat is not too dense, or has high flammable inclusion content, or at high wind speeds (Yokelson et al., 1996; 1997).

During eight days from 31 October through 7 November, we sampled 35 separate plumes at six different peatland areas with two areas being revisited. All smoke sampling was conducted directly in the visible plumes (Fig. S1) and all background sampling was conducted just outside (usually upwind) of the plumes in paired fashion. The surface fuels at all sites were non-existent or limited to ferns, charred logs, or patchy second growth forest, but they were neither present in heavy loading nor burning in most cases. This facilitated sampling "pure emissions" from the smoldering peat. On each day from 1-7 November, about four plumes originating from various peat types or depths were grab sampled about ten times each by FTIR, at least once by WAS, and usually by filters. This provided data for 27 plumes each assigned a letter identifier in our tables from A-Z-AA. Eight additional plumes were quickly, opportunistically, sampled by just WAS, which was the fastest sampling method to complete. On 5 and 6 November, seven of the plumes with letter identifiers were also sampled continuously between 10-30 minutes apiece with both PAXs (coincident with FTIR, WAS, and filter sampling). Twenty-two filter samples were collected from 19 different "lettered" plumes from 1-7 November. The full set of filter based analyses will be reported separately





(Jayarathne et al., 2016). The sites and fires sampled included a variety of peat types, disturbance levels, spread modes, burn depths, etc. A brief chronological narrative of the sampling follows and most of the site characteristics that we were able to document are shown in Table S1.

*31 October (Site 1).* Two WAS samples were collected while scouting this site known locally as "South Bridge

West" late in the afternoon. The site (Site #1 in Tab. S1) had hemic and fibric peat burning at 30-60 cm depth and was the most disturbed of all the sites sampled.

*1 November (Site 1), plumes A-D.* The "South Bridge West" site #1 was revisited and sampled by WAS, FTIR, and filters, which began the series of intensively sampled plumes designated by letters. Plume C included emissions from surface peat that were partially impacted by flames during wind gusts.

*2 November (Site 2), plumes E-H.* This site (#2) was the least disturbed of the sites we sampled, but had been logged and was known to have burned once before the fire we sampled. In addition, site #2 was close enough to a canal that its hydrology would have been impacted. The site is known locally as "South Bridge East." The peat was hemic and fibric and burn depth ranged from 18-28 cm.

*3 November (Site 3), plumes I-L.* The "White Shark (Hiu Putih)" site comprised hemic and fibric peat burning at

depths of 33-52 cm.

*4 November: (Site 4) plumes M-N; (Site 5) Plume P.* Site 4 was known locally as the "Mahir Mahar" site and plume M provided our best measurements of the emissions from burning sapric peat. The other plumes sampled were burning in hemic and fibric peat types. The burn depths sampled on this day varied over a narrow range near 21-22 cm.

*5 November (Site 1), plumes Q-T.* The South Bridge West site was revisited. Burn depths were 25-50 cm and the peat was hemic and fibric.

*6 November (Site 2), plumes U-W.* The South Bridge East site was revisited. The peat was hemic and fibric and burn depths were 20-30 cm.

*7 November (Site 6), plumes X-Z-AA.* Some shallow peat combustion was sampled at this site, known locally as

Tangkiling Road.

## 2.2 Instrument descriptions and calculations

### 2.2.1 Land-based Fourier transform infrared (LA-FTIR) spectrometer

A rugged, cart-based, mobile FTIR (Midac, Corp., Westfield, MA) designed to access remote sampling locations was used for trace gas measurements (Christian et al., 2007). We note for other researchers that the soft peat surface

was not easily traversed with the rolling cart, which usually had to be carried. In addition, all equipment was protected from underlying ash and dust with a tarp. The vibration-isolated optical bench consists of a Midac spectrometer with a Stirling cycle cooled mercury-cadmium-telluride (MCT) detector (Ricor, Inc.) interfaced with a closed multipass White cell (Infrared Analysis, Inc.) that is coated with a halocarbon wax (1500 Grade, Halocarbon Products Corp.) to minimize surface losses (Yokelson et al., 2003). In the grab sampling mode air samples are

drawn into the cell by a downstream pump through several meters of 0.635 cm o.d. corrugated Teflon tubing. The air samples are then trapped in the closed cell by Teflon valves and held for several minutes for signal averaging to



increase sensitivity. Once the IR spectra of a grab sample are logged with cell temperature and pressure (Minco TT176 RTD, MKS Baratron 722A) on the system computer, a new grab sample can be obtained resulting in many grab samples for each peat fire smoke plume and "paired" backgrounds. Spectra were collected at a resolution of 0.50 cm$^{-1}$ covering a frequency range of 600-4200 cm$^{-1}$. Since some other recent reports of the use of this system

(Akagi et al., 2013), several upgrades have been made: (1) addition of a retroreflector to the White cell mirrors increased the optical pathlength from 11 m to 17.2 m, lowering previous instrument detection limits, (2) replacing the Teflon cell coating with halocarbon wax to enable measurements of ammonia ($NH_3$), hydrogen chloride (HCl), and other species prone to adsorption on surfaces, (3) mounting the mirrors to a stable carriage rather than the previous method of gluing them to the cell walls, (4) the above mentioned Stirling cycle detector, which gave the

same performance as a liquid-nitrogen-cooled detector without the need for cryogens, (5) the addition of two logged flow meters (APEX, Inc.) and filter holders to enable the system to collect particulate matter on Teflon and quartz filters for subsequent laboratory analyses. The new lower detection limits vary by gas from less than 1 ppb to ~100 ppb, but are more than sufficient for near-source ground-based sampling since concentrations are much higher (e.g. ppm range) than in lofted smoke (Burling et al., 2011). Gas-phase species including carbon dioxide ($CO_2$), carbon

monoxide (CO), methane ($CH_4$), acetylene ($C_2H_2$), ethylene ($C_2H_4$), propylene ($C_3H_6$), formaldehyde (HCHO), formic acid (HCOOH), methanol ($CH_3OH$), acetic acid ($CH_3COOH$), furan ($C_4H_4O$), hydroxyacetone ($C_3H_6O_2$), phenol ($C_6H_5OH$), 1,3-butadiene ($C_4H_6$), nitric oxide (NO), nitrogen dioxide ($NO_2$), nitrous acid (HONO), $NH_3$, hydrogen cyanide (HCN), hydrogen chloride (HCl), sulfur dioxide ($SO_2$) were quantified by fitting selected regions of the mid-IR transmission spectra with a synthetic calibration non-linear least-squares method (Griffith, 1996;

Yokelson et al., 2007). A few species were sometimes not above the detection limit in background air, but are retrieved from absorption spectra made from smoke/background so the excess amounts are inherently returned. $SO_2$ and $NO_2$ were not observed above the detection limit in the background or the most concentrated smoke and are not discussed further. An upper limit 1σ uncertainty for most mixing ratios is ±10%. Pre-mission calibrations with NIST-traceable standards indicated that CO, $CO_2$, and $CH_4$ had an uncertainty between 1-2%, suggesting an upper

limit on the field measurement uncertainties for CO, $CO_2$, and $CH_4$ of 3-5%. The $NO_x$ species have the highest interference from water lines under the humid conditions in Borneo and the uncertainty for NO is ~25%.

In addition to the primary grab sample mode, the FTIR system was also used in a real-time mode to support the PAX (vide infra) and filter sampling when grab samples were not being obtained. Side by side Teflon and quartz filter holders preceded by cyclones to reject particles with an aerodynamic diameter > 2.5 microns were followed by

logged flow meters. The flow exiting the meters was then combined and directed to the multipass cell where IR spectra were recorded at ~1.1 second time resolution. The PAX sample line was co-located with the filter inlet and sampled in parallel from the same location. In real-time filter/PAX mode we did not employ signal averaging of multiple FTIR scans and the signal to noise is lower at high time resolution. In addition, there could be sampling losses of sticky species such as $NH_3$ on the filters so we did not analyze the real-time data for these species.

However, the data quality was still excellent for $CO_2$, CO, and $CH_4$. This allowed the time-integrated particle mass and PAX signals to be compared to the simultaneously measured time-integrated mass of the three gases most





needed for EF calculations (Sect 2.3) and provided additional measurements of the emissions for these three gases as described in detail in the filter sampling companion paper (Jayarathne et al., 2016).

### 2.2.2 Whole air sampling (WAS) in canisters

Whole air samples were collected in evacuated 2 L stainless steel canisters equipped with a bellows valve that were pre-conditioned by pump-and-flush procedures (Simpson et al., 2006). The canisters were filled to ambient pressure directly in plumes or adjacent background air to enable subsequent measurement and analysis of a large number of gases at the University of California, Irvine. Species quantified included $CO_2$, CO, $CH_4$ and up to 100 non-methane organic gases by gas chromatography (GC) coupled with flame ionization detection, electron capture detection, and quadrupole mass spectrometer detection as discussed in greater detail by Simpson et al. (2011). Typically ~70 of the NMOGs were enhanced in the source plumes and we do not report the results for most multiply-halogenated species, which are generally not emitted by combustion (Simpson et al., 2011). We also do not report the higher-chain alkyl nitrates, which are often secondary photochemical products and were not enhanced in these fresh peat fire plumes. Peaks of interest in the chromatograms were individually inspected and manually integrated. The limit of detection for most NMOGs was less than 20 pptv, well below the concentrations that were sampled. Styrene is known to decay in canisters and the styrene data may be lower limits.

### 2.2.3 Photoacoustic extinctiometers (PAX) at 405 nm and 870 nm

Particle absorption and scattering coefficients ($B_{abs}$, $B_{scat}$), single scattering albedo (SSA), and absorption Ångström exponent (AAE) at 405 nm and 870 nm were measured directly at 1 s time resolution using two photoacoustic extinctiometers (PAX, Droplet Measurement Technologies, Inc., CO). This monitored the real-time absorption and scattering resulting from BC and (indirectly) BrC. The two units were mounted with a common inlet, desiccator (Silica Gel), and gas scrubber (Purafil) in rugged, shock-mounted, Pelican military-style hard cases. Air samples were drawn in through conductive tubing to 1.0 µm size-cutoff cyclones (URG) at 1 L/min. The continuously sampled air was split between a nephelometer and photoacoustic resonator enabling simultaneous measurements of scattering and absorption at high time resolution. Once drawn into the acoustic section, modulated laser radiation was passed through the aerosol stream and absorbed by particles in the sample of air. The energy of the absorbed radiation was transferred to the surrounding air as heat and the resulting pressure changes were detected by a sensitive microphone. Scattering coefficients at each wavelength were measured by a wide-angle integrating reciprocal nephelometer, using photodiodes to detect the scattering of the laser light. The estimated uncertainty in PAX absorption and scattering measurements has been estimated as ~4-11% (Nakayama et al., 2015). Additional details on the PAX instrument can be found elsewhere (Arnott et al., 2006; Nakayama et al., 2015). For logistics reasons it was only practical to sample fresh peat fire plumes with the PAXs on two days.

Calibrations of the two PAXs were performed during the deployment using the manufacturer recommended absorption and scattering calibration procedures utilizing ammonium sulfate particles and a kerosene lamp to generate pure scattering and strongly absorbing aerosols, respectively. The calibrations of scattering and absorption of light were directly compared to measured extinction by applying the Beer-Lambert Law to laser intensity





attenuation in the optical cavity (Arnott et al., 2000). As a quality control measure, we frequently compared the measured total light extinction ($B_{abs}$ + $B_{scat}$) to the independently measured laser attenuation. For nearly all the 1 s data checked, the agreement was within 10% with no statistically significant bias, consistent with (though not proof of) the error estimates in Nakayama et al. (2015). Finally, after the mission a factory measurement of the 405 nm

absorption in the PAX was performed with $NO_2$ gas that was within 1% of the expected result (Nakayama et al., 2015). As part of this factory calibration, to account for the $NO_2$ quantum yield, the laser wavelength was precisely measured as 401 nm. This difference from the nominal 405 nm wavelength is common and we continue to refer to the wavelength as 405 nm since this is a standard nominal wavelength for aerosol optical measurements. This impacts the calculated values for AAE by only 0.3 % and the absorption attribution by 1.0 % (Sect. 2.3).

**2.2.4 Other measurements**

Peat samples were collected just ahead of the burning front for fuel moisture measurements. A brief description of the filter collection process is given here and the details of the post-mission analyses will be described elsewhere (Jayarathne et al., 2016).

**2.2.5 PM$_{2.5}$ filter collection for offline analysis**

PM$_{2.5}$ was collected through 0.635 cm o.d. Cu tubing and PM$_{2.5}$ cyclones onto pre-weighed 47 mm Teflon filters and pre-cleaned 47 mm quartz fiber filters (QFF) (PALL, Life Sciences, Port Washington, NY) in both smoke plumes and directly-upwind background air. QFF were pre-baked at 550˚C for 18 hours before sampling to remove contaminants and stored in cleaned, aluminum foil-lined petri dishes sealed with Teflon tape.

*PM$_{2.5}$ mass measurements.* Before and after sample collection Teflon filters were conditioned for 48 hours in a
desiccator and weighed using an analytical microbalance (Mettler Toledo XP26) in a temperature and humidity controlled room. Particulate mass (PM) was calculated from the difference between pre-and post-sampling filter weights, which were determined in triplicate. PM per filter was converted to mass concentration using the sampled air volume. Uncertainty in the excess mass in the smoke plumes was propagated using the standard deviation of the smoke PM, the standard deviation of the background PM, and 10 % of the PM concentration.

*EC OC analysis.* EC and OC were measured by thermal optical analysis (Sunset Laboratory, Forest Grove, OR) following the NIOSH 5040 method (NIOSH, 1999) using 1.00 cm$^2$ sub-samples of the quartz fiber filters. The EC/OC split was determined by thermal optical transmittance (TOT). The OC and EC concentrations ($\mu g\ m^{-3}$) were calculated using the total filter area and the sampled air volume. The OC uncertainty was propagated using the standard deviation of the field blanks, the standard deviation of background filters, and 10 % of the OC
concentration. Instrumental uncertainty (0.05 $\mu g\ cm^{-2}$), 5 % of the EC concentration, and 5 % of the measured pyrolyzed carbon concentration were used to propagate EC uncertainty.

*Back-up filter collection.* In order to assess the positive sampling artifacts from carbonaceous gas adsorption, a second QFF (back-up) filter was placed following the first QFF (front) filter. These QFF filters were analyzed for EC and OC as described previously. EC was not detected on any of the back-up filters. On average, the OC
concentration on backup filters was 4.8 % of OC on front filters. At the high concentrations sampled both QFF





would saturate with respect to gas adsorption indicating that ~5 % of the front filter OC was due to positive sampling artifacts (Kirchstetter et al., 2001).

*Background filter collection.* In order to correct for ambient background $PM_{2.5}$, background filter samples were collected in background air outside, but adjacent to the smoke plumes for 20 minutes (similar to the smoke sampling
times). These filters were also analyzed for $PM_{2.5}$ mass, EC and OC as described above. EC was not detected on any of the background filters, while OC levels were consistent with gas adsorption described previously. The backgrounds were very similar and on average, the background contributed 0.60 % of $PM_{2.5}$ mass, indicating that background contributions to PM mass were very minor in relation to the peat burning smoke. Nonetheless, the average background value was subtracted from the smoke samples during data workup to calculate the contributions
from the smoke plumes.

**2.3 Emission ratio and emission factor determination**

The excess mixing ratios above the background level (denoted $\Delta X$ for each gas-phase species "X") were calculated for all the gas-phase species in the grab samples and $CO_2$, CO, and $CH_4$ in the real-time data. The grab samples were collected in a way that avoided possible artifacts for some gases due to adsorption on filters or in flow meters and
they were used to produce a self-consistent complete set of data on trace gas emissions as described next. The molar emission ratio (ER, e.g. $\Delta X/\Delta CO$) for each gaseous species X relative to CO or $CO_2$ was calculated for all the FTIR and WAS species. The plume-average ER for each FTIR or WAS species measured in multiple grab samples was estimated from the slope of the linear least-squares line (with the intercept forced to zero) when plotting $\Delta X$ versus $\Delta CO$ (or $\Delta CO_2$) for all samples of the source (Yokelson et al., 2009; Christian et al., 2010; Simpson et al., 2011).
Forcing the intercept decreases the weight of the lower points relative to those obtained at higher concentrations that reflect more emissions and have greater signal to noise. Alternate data reduction methods usually have little effect on the results as discussed elsewhere (Yokelson et al., 1999). For a handful of species measured by both FTIR and WAS it is possible to average the ERs from each instrument for a source together as in Yokelson et al. (2009). However, in this study, we either worked up the independently sampled WAS data as a separate set of ER or used
the more extensive FTIR ERs when there were a few "overlap species" (primarily $CH_3OH$, $C_2H_4$, $C_2H_2$, and $CH_4$). From the ERs, emission factors (EFs) were derived in units of grams of species X emitted per kilogram of dry biomass burned by the carbon mass balance method, which assumes all of major carbon-containing emissions have been measured (Ward and Radke, 1993; Yokelson et al., 1996, 1999):

$$EF(X)\left(g\,kg^{-1}\right) = F_C \times 1000 \times \frac{MM_x}{AM_C} \times \frac{\dfrac{\Delta X}{\Delta CO}}{\sum_{j=1}^{n}\left(NC_j \times \dfrac{\Delta C_j}{\Delta CO}\right)} \qquad (1)$$

where $F_C$ is the measured carbon mass fraction of the fuel; $MM_x$ is the molar mass of species X; $AM_C$ is the atomic mass of carbon (12 g $mol^{-1}$); $NC_j$ is the number of carbon atoms in species j; $\Delta C_j$ or $\Delta X$ referenced to $\Delta CO$ are the fire-average molar emission ratios for the respective species. The carbon fraction was measured (ALS Analytics, Tucson) for seven samples of Kalimantan peat from sites ranging from heavy to no disturbance and averaged 0.5793





± 0.0252 (Stockwell et al., 2014). EFs are proportional to assumed carbon content, making future adjustments to evolving literature-average EFs trivial if warranted based on additional carbon content measurements. The denominator of the last term in Eqn. (1) estimates total carbon. For nearly all the plumes, the mass ratio of EC and OC to the simultaneous co-located CO, measured by the FTIR (see below), was added to the estimate of total

carbon. Thus, our total carbon estimate for the grab samples includes all the gases measured by the FTIR or WAS in grab samples of a source and the carbon in the aerosol measured on the filters. Ignoring the carbon emissions not included or not measureable by our suite of instrumentation (typically higher molecular weight oxygenated organic gases) likely inflates the EF estimates by less than ~1-2 % (Yokelson et al., 2013; Stockwell et al., 2015), which is small compared to the 4% uncertainty due to natural variability in peat carbon content.

Biomass fire emissions vary naturally as the mix of combustion processes varies. The relative amount of smoldering and flaming combustion during a fire can be roughly estimated from the modified combustion efficiency (MCE). MCE is defined as the ratio $\Delta CO_2/(\Delta CO_2+\Delta CO)$ and is mathematically equivalent to $(1/(1+\Delta CO/\Delta CO_2)$ (Yokelson et al., 1996). Flaming and smoldering combustion often occur simultaneously during biomass fires, but a very high MCE (~0.99) designates nearly pure flaming (more complete oxidation) while a lower MCE (~0.75-0.84 for

biomass fuels) designates pure smoldering. Plume-average MCE was computed for all plumes using the plume average $\Delta CO/\Delta CO_2$ ratio as above. In the context of biomass or other solid fuels, smoldering refers to a mix of solid-fuel pyrolysis (producing NMOG and OA) and gasification (producing mainly $NH_3$, $CH_4$, and inorganic gases with little visible aerosol) (Yokelson et al., 1997).

The time-integrated excess $B_{abs}$ and $B_{scat}$ from the PAXs were used to directly calculate the plume average single

scattering albedo (SSA, defined as $B_{scat}/(B_{scat} + B_{abs})$) at both 870 and 405 nm for each source. The PAX time-integrated excess $B_{abs}$ at 870 and 405 was used directly to calculate each plume-average absorption Ångström exponent (AAE, Eqn. 2).

$$AAE = -\frac{\log\left(\frac{B_{abs,1}}{B_{abs,2}}\right)}{\log\left(\frac{\lambda_1}{\lambda_2}\right)} \qquad (2)$$

Aerosol absorption is a key parameter in climate models, however, inferring absorption from total attenuation of

light by particles trapped on a filter, or from the assumed optical properties of a mass measured by thermal/optical processing, incandescence, etc. can sometimes suffer from artifacts (Andreae and Gelencsér, 2006; Subramanian et al., 2007). In the PAX, the 870 nm laser is absorbed in-situ by black carbon containing particles only, without filter or filter-loading effects that can be difficult to correct. We directly measured aerosol absorption ($B_{abs}$, $Mm^{-1}$) and used the literature-recommended mass absorption coefficient (MAC) (4.74 $m^2/g$ at 870 nm) to estimate the BC

concentration ($\mu g/m^3$) (Bond and Bergstrom, 2006). The PAXs (and filters) were co-sampled with the FTIR measuring $CO_2$, CO, and $CH_4$ in real-time. The mass ratio of the integrated excess BC in the plume measured on the PAX to the integrated excess CO measured by the FTIR was multiplied by the EF CO based on the real-time FTIR data to determine EFs for BC (g/kg). Note the total C for the carbon mass balance for the EFs calculated for real-time data is based on the integrated excess amounts of just the three main gases and aerosol carbon, which will





inflate the EFs by a small amount (typically 1-3 %) compared to the larger suite of gases used for the grab sample calculations.

To a good approximation, $sp^2$-hybridized carbon (i.e. BC) has an AAE of $1.0 \pm 0.2$ and absorbs light proportional to frequency. Thus, $B_{abs}$ due only to BC at 405 nm would be expected to equal $2.148 \times B_{abs}$ at 870 nm and we assumed that excess absorption at 405 nm, above the projected amount, is associated with BrC absorption. This method of attributing BrC absorption is based on several assumptions discussed in detail elsewhere that are likely most valid in cases where the BrC absorption is dominant such as in these peat fire smoke plumes (Lack and Langridge 2013). In theory, a BrC concentration ($\mu g/m^3$) could be calculated using a literature-recommended BrC MAC of 0.98 $m^2/g$ at 404 nm (Lack and Langridge, 2013). The BrC mass calculated this way would be intended to be roughly equivalent to the total OA mass, which as a whole weakly absorbs UV light, and not the mass of the actual chromophores. However, the MAC of Lack and Langridge (2013) is appropriate for more typical biomass burning with a mix of flaming and smoldering, whereas the peat aerosol is overwhelmingly organic and at low BC/OA ratios the MAC is much smaller (Saleh et al., 2014; Olson et al., 2015). Thus, instead we divided the $B_{abs}$ at 405 nm assigned to BrC by the co-measured OC mass to estimate the peat smoke MAC referenced to bulk OC. The EFs for scattering and absorption at 870 and 405 nm (EF $B_{abs}$, EF $B_{scat}$) are reported directly in units of $m^2$ per kg of dry fuel burned by multiplying the ratios of $B_{abs}$ and $B_{scat}$ to co-measured real-time CO by the real-time EF CO. We note that most of the related measurements of elemental and organic carbon on the filters will be discussed separately by Jayarathne et al. (2016).

### 3 Results and Discussion

#### 3.1 Trace gas emission factors

In general, we found very high correlation in the ER plots indicating the plumes were well-mixed and implying low uncertainty in the individual plume EFs. Figure 1 shows a selection of such plots for plume N and it is also seen that the smoke mixing ratios were far above background. There were nine instances when the same gas was measured by both WAS and FTIR in nearly the same place and seven of these nine cases agree within the combined uncertainty. The other two cases are less close, but this experiment was not well-designed for comparison. We have noted excellent WAS/FTIR agreement previously under more rigorous, but drier conditions (Christian et al., 2003) and we found that these 2015 field WAS results compared well with on-line measurements during FLAME-4 peat fire sampling for many major species as discussed later in the paper.

Table S2 presents all the trace gas EFs for all 35 plumes sampled while Table 1 shows all our study-average EFs and one standard deviation of the means for all the gases that were significantly elevated in the smoke plumes. In the pure peat combustion that we were able to sample, the major trace gas emissions by mass (EF > ~0.5 g/kg) were: carbon dioxide (1564 ± 77), carbon monoxide (291 ± 49), methane (9.51 ± 4.74), hydrogen cyanide (5.75 ± 1.60), acetic acid (3.89 ± 1.65), ammonia (2.86 ± 1.00), methanol (2.14 ± 1.22), ethane (1.52 ± 0.66), dihydrogen (1.22 ± 1.01), propylene (1.07 ± 0.531), propane (0.989 ± 0.644), ethylene (0.961 ± 0.528), benzene (0.954 ± 0.394), formaldehyde (0.867 ± 0.479), hydroxyacetone (0.860 ± 0.433), furan (0.772 ± 0.035), acetaldehyde (0.697 ±



0.460), and acetone (0.691 ± 0.356). These results are shown in a bar chart in Fig. 2. $C_6$-$C_{10}$ alkanes summed to 0.87 ± 0.57 g/kg, which overlaps with the 0.59 g/kg of $C_6$-$C_{10}$ alkanes emitted by a peat fire sampled by two-dimensional gas chromatography in the FLAME-4 lab study (Hatch et al., 2015). Hatch et al. (2015) also measured 0.43 g/kg of $C_{11}$-$C_{15}$ alkanes, which is probably a reasonable estimate for our field fires. The larger alkanes (> $C_{10}$) are efficient

OA precursors (Presto et al., 2010). BTEX (benzene, toluene, ethylbenzene, xylenes) compounds are also high-yield OA precursors (Wang et al., 2014) and important air toxics; they were emitted in total at 1.49 ± 0.64 g/kg. Air toxics are discussed further in Sect 3.5.2 with the FLAME-4 lab data included. Additional discussion of NMOG emissions and detailed comparison with previous (e.g. FLAME-4) trace gas measurements on lab peat fires is presented in Section 3.5.1.

The modified combustion efficiency (MCE) of the smoke sources ranged from 0.693 to 0.835 with an average of 0.772 ± 0.035 ($n$ = 35) indicating essentially pure smoldering combustion. For most biomass fires there is both flaming and smoldering and so EFs correlate with MCE, but these fires burned by smoldering only with no high MCE values (e.g. >0.9) and little or no correlation of EFs with MCE. It is important to consider if EFs are related to peat characteristics, especially peat characteristics that could be mapped. However, given our sample size and some

mixing of peat types by the disturbance regimes, we have not attempted such an analysis yet.

### 3.2 Aerosol optical properties and emission factors

Figure 3 shows an example of the PAX real-time $B_{abs}$ at 870 and 405 nm collected on 5 Nov along with the co-located CO data. Note the scaling of the axes and the dominance of $B_{abs}$ at 405 nm, though the ratio of 870/405 is seen to increase towards the end of the sampling period (the traces are slightly offset so that the background trace is

visible.) The excess values above background that were used to calculate all the quantities described above had similar excellent signal to noise in all cases.

Table 2 shows all PAX-measured quantities, the MCE from the co-sampled real-time FTIR data, and the small subset of filter EC, OC, and $PM_{2.5}$ data that were co-sampled with the PAXs for all 7 plumes along with the study averages and standard deviations. Consistent with the lack of flaming, the emissions of BC were negligible (0.0055

± 0.0016 g/kg) (Christian et al., 2003; Liu et al., 2014). Aerosol absorption at 405 nm was 52 times larger than at 870 nm and BrC contributed an estimated 96% of the absorption at 405 nm. Average AAE was 4.97 ± 0.65 (range 4.29-6.23). The SSA at 405 nm (0.974 ± 0.016, range 0.941-0.989) was marginally lower than SSA at 870 nm (0.998 ± 0.001, range 0.997-0.999). Clearly, estimating aerosol absorption from BC measurements alone would be inadequate for this source.

Pure pyrolysis has lower MCE than glowing and thus, pyrolysis is implicated as the source of BrC via the correlation of AAE with lower MCE ($r^2$ = 0.65) (Fig. 4a). We note the data cover a small MCE range and thus the relationship shown is not well constrained for extrapolation much beyond the range shown. We also find that AAE correlates strongly with SSA at 405 nm (Fig. 4b). In this case, the trend line shown is likely illustrative of peat fire aerosol, but again, not suitable for extrapolation to other fuels or beyond the range shown.

By plotting EF $B_{scat}$ versus EF $PM_{2.5}$ for all seven plumes sampled by PAX and filters (Fig. 5) we get a rough estimate of the mass-scattering efficiency (MSE) of the peat fire aerosol at 405 nm based on the slope of 2.96 ± 0.67



$m^2/g$ ($r^2 = 0.80$). The plot compared EFs measured in the same plumes, but in some cases at slightly different times due to a PAX auto-zero or a filter clogging. If we restrict the plot to the four plumes where the timing of the sampling was identical, the slope is 3.05 $m^2/g$ ($r^2 = 0.81$). Either value of the MSE is close to MSEs obtained closer to 500 nm ($3 - 5$ $m^2/g$) in other studies of biomass burning aerosol with lower values characteristic of fresher smoke

(Tangren, 1982; Patterson and McMahon, 1984; Nance et al., 1993; Burling et al., 2011). However, based on average BB aerosol size distributions (Reid et al., 2005), our MSE may be underestimated on the order of 5-10% due to the difference in sampling cut-offs (2.5 microns for filters and 1.0 microns for PAX). By comparing the EF $B_{abs}$ at 405 nm assigned to BrC with EF OC from the filters on the same plumes (Fig. 6) we can estimate the mass absorption coefficient (MAC) of the bulk OC. As above, two MAC estimates are possible. Using the mean value for

all 7 plumes we get $0.09 \pm 0.08$ $m^2/g$ where the large coefficient of variation is due to one larger MAC value near 0.27 $m^2/g$. If instead we plot EF $B_{abs-405}$ versus EF OC just for the four plumes sampled over the exact same time period (but different size cutoffs; blue points in Fig. 6) we get a slope of $0.071 \pm 0.03$ $m^2/g$. The MACs obtained either way are similar, but again underestimated by a few percent due to cutoff differences and much smaller than MACs for average biomass burning OA (0.98; Lack and Langridge, 2013). However, we confirm the expected

MAC near 0.1 $m^2/g$ for the extremely low BC (or EC) to OA ratio in the aerosol (Saleh et al., 2014; Olson et al., 2015).

While EC and BC are considered approximately equivalent for some combustion sources (e.g. diesel fuel combustion), our EF EC for peat fires is noticeably larger than the EF BC although both EC and BC values are very small (Table 2) compared to typical values for combustion aerosol. This is the expected result in this case for several

reasons. The peat smoke plumes sampled outdoors likely contain very small amounts of soot from rare instances of flaming and also a small amount of entrained small char particles produced by pyrolysis of the peat on site by the glowing combustion front (Santín et al., 2016). Both soot and char are detected to some extent as EC (Andreae and Gelencsér, 2006; Han et al., 2007; 2010; 2016) and our EC sub-fractions evolving at lower temperatures confirm some char was present (NIOSH, 1999). The char particles tend to be larger (1-100 microns, Han et al., 2010) and

would be more efficiently sampled by the filters, which had a 2.5 micron cut-off as opposed to the PAX with a 1.0 micron cut-off. Char tends to absorb long wavelengths less efficiently than soot (Han et al., 2010) and the PAX would therefore be relatively insensitive to any sampled char for this reason also. The accuracy of both the PAX and the thermal optical EC detection is challenged by the low EC or BC to OC ratio (Andreae and Gelencsér, 2006). Yet, both measurements are useful and point to the same key results: that the aerosol is overwhelmingly organic and the

organic fraction contributes most of the light absorption.

In a previous study of aerosol emissions from burning Sumatran peat in a lab setting, Christian et al. (2003) measured an EF for OC + EC by the thermal optical technique of ~6 g/kg that had OC/EC of 151. More extensive comparison of our field $PM_{2.5}$, EC, and OC data with lab measurements, including the FLAME-4 EC/OC data will be presented in Jayarathne et al. (2016).

Turning to optical properties, Liu et al. (2014) reported some SSA values and the AAE for smoldering Kalimantan peat (Fire 114) from FLAME-4: MCE (0.74), SSA 405 (0.94), SSA 781 (1.00), and AAE (6.06). These are very consistent with our data (Table 2) and especially with our lowest MCE field sample: MCE (0.726), SSA 405




(0.941), SSA 870 (0.997), and AAE (6.23). They also report data for a FLAME-4 peat fire with some brief flaming (Fire 154) and obtain for example an AAE of 3.02, which is below our lowest AAE of 4.28. Their average AAE 4.45 ± 2.19 for Indonesian peat is not significantly smaller than ours (4.97 ± 0.65) and it should be kept in mind that the determination and comparisons of AAE can be affected by the use of different wavelength pairs (Lewis et al., 2008; Chakbarty et al., 2016). In summary, when comparing to published laboratory studies of tropical peat burning, especially for smoldering combustion in the lab, we get good agreement in the sense of extremely low EC or BC to OC ratios and for the aerosol optical properties.

### 3.3 Representativeness and comparison to other field studies

The biomass of the surface layer in logged/disturbed peatlands is small compared to the peat, and even the biomass of intact peat-swamp forest is small compared to peat loading as noted by Page et al. (2002). However, peat is only one component of the total peatland fuel and potentially a diminishing component as exploitation and repeated fires are continued over many years (Konecny et al., 2016). As the peat fuels are consumed on a site, the loading of surface fuels likely also decreases. We did not see much evidence of active surface fuel combustion, but our sampling was just after the peak regional $PM_{10}$ levels, which may have had a larger contribution from surface fuels. Numerous "hotspots" were detected in the region and both flaming and smoldering were evident in the news media coverage (https://worldview.earthdata.nasa.gov/). The fraction of total, annual regional emissions due to emissions generated during the peak regional impacts should not be overestimated since a long period of moderately elevated emissions could produce as much or more emissions as a shorter, higher level of emissions. The overall mix of fuels burning in the region during the peak regional pollution would have been hard to assess in any case since visibility dropped to ~10 m making driving dangerous and even a regional fire survey with an aircraft problematic. Further, surface fuel emissions would likely be associated with some amount of flaming combustion that would be hard to sample properly with most ground-based instruments. Finally, under extremely polluted conditions it is hard to acquire background samples or isolate and measure individual fuel contributions/EFs so that the variable relative contributions of peat and surface fuels (primary and secondary forest, cropland, grassland, etc.) can be explicitly modeled on a regional scale. Our sampling, somewhat fortuitously, unambiguously probed the emissions from the major fuel component, peat, of special concern in Southeast Asia.

Our sampling was also near the end of the fire season when the relative amount of total annual deep burning versus total annual surface burning could potentially be measured (an earlier assessment would underestimate the deep peat burning). We sampled and observed areas with peat burning at depths from 18 to 60 cm. However, we also accessed our sites at times across areas that had recently burned with consumption of some surface fuels, but with only shallow consumption of the organic soil layer. Thus, applying an average peat burn depth for all burned area from our sampled burn depths would be biased high and a better estimate of the average burn depth will likely result from the Lidar data collected. There were significant areas where a deep burn depth is clearly not accurate. On the other hand, burned area is likely underestimated in inventories since they rely on remote sensing data that misses hotspots, burned area, and the fire products used in top-down approaches. This is due to high regional cloud cover; orbital gaps; rapid green-up, which is strongly associated with shallow burn depth (Cypert, 1961; Kotze, 2013); and other





factors (Reid et al., 2013). Thus, overestimating burn depth and underestimating burned area tend to cancel. A 2015 airborne campaign surveying regional smoke could have theoretically assessed the overall regional smoke characteristics, but did not occur. With the caveat that fire use has evolved in Kalimantan over the years, we can compare to airborne atmospheric chemistry measurements conducted during the 1997 El Niño haze event as detailed

next.

We now compare our ground-based measurements of "pure" peat smoke to the only available airborne regional smoke measurements, which were part of the Pacific Atmospheric Chemistry Experiment 5 (PACE-5) campaign in Kalimantan during the peak of another El Niño event (Sawa et al., 1999). During late October 1997, airborne sampling was conducted west of Banjarmasin along a flight leg several hundred km long at four flight levels

between 1.3 and 4.4 km altitude. The flight was ~100 km south of Palangkaraya and encountered 3-9 ppm of CO and ~500 m visibility at lower altitudes (Sawa et al., 1999). Gras et al. (1999) noted that no visible flame fronts were observed from the aircraft and estimated one SSA for a Kalimantan smoke plume as 0.98. This is close to our 530 nm value if we interpolate between 870 and 405 nm (0.981). They measured large hygroscopic growth factors of 1.65 which agreed well with tests of peat combustion they cite by Golitsyn et al. (1988). From the same flight Sawa

et al. (1999) reported $NO_x/CO$ ERs of 0.00019 to 0.00045, which they attributed to a lack of flaming combustion, but also possibly faster losses of $NO_x$ than CO. We observed several individual values in their range (our minimum was 0.00028), but our average $NO_x/CO$ ER is higher ($0.0012 \pm 0.0007$). The comparison is good in that the ranges overlap and are consistent with smoldering combustion, but some fast $NO_x$ losses probably also impacted the airborne ERs. The PACE-5 team speculated that high $SO_2$ emissions could contribute to the hygroscopicity and cited

unpublished lab tests that confirmed high $SO_2$ from burning peat. We did not see evidence of elevated $SO_2$, but our measurements were conducted further inland, possibly away from Holocene coastal sulfidic sediments invoked by Gras et al. (1999) as a possible source of $SO_2$. During FLAME-4, no $SO_2$ was detected from burning peat in the lab except for the one sample of coastal peat which was collected in North Carolina (Table S2 in Stockwell et al., 2015). This suggests that the emissions from burning coastal peat deposits are impacted by their known chemical

differences (Cohen and Stack, 1996).

Hamada et al. (2013) measured $CO_2$, CO, and $CH_4$ emissions from a peat fire near Palangkaraya during the 2009 El Niño. Based on 23 samples, they report $CO/CO_2$ and $CH_4/CO_2$ ERs of 0.382 and 0.0261; 31 and 56% higher than our study averages, respectively, but within our range for individual plume averages. Their data are consistent with a smoldering-dominated burn and an MCE of 0.724, which is within our range for individual fires (one of ours was

lower (0.693), though our study average was higher ($0.772 \pm 0.035$).

### 3.4 Application of emission factors

The basic application of emission factors is to multiply them by a total fuel consumption to generate total emissions for a desired region (Seiler and Crutzen, 1980). Our EFs in this work are intended for use with peat consumption estimates to calculate total emissions from the peat component. Major uncertainties would include natural variation

of the EFs (e.g. the standard deviations of the EFs given in Table 2) and variation in %C, density, and burn depth of the peat. Konecny et al. (2016) list some other %C and burn depth measurements, which are generally close to our





values. We plan to present further data on these issues in a separate paper. We note that in a previous review of BB EFs Akagi et al. (2011) estimated literature average values for EFs for pure peat. Following Page et al. (2002) they also computed "peatland" EFs by combining the peat EFs and fuel consumption with EFs and fuel consumption for tropical peat swamp forest, which was considered as the only surface fuel type. This was potentially appropriate for

1997. However, given on-going land-use trajectories, it is now clear that many different types of surface fuels and a variety of fuel combinations are important (Miettinen et al., 2016). The work here presents EFs specific for the major peat component that can be coupled with peat fuel consumption estimates and that ideally contribute to emissions estimates after combining with fuel consumption estimates and EFs for the relevant surface fuel types. Many of the EFs and fuel consumption values for other surface fuel types are tabulated in Akagi et al., (2011).

Another earlier set of trace gas EF previously available for tropical peat burning was from a laboratory study (Christian et al., 2003) and was also adopted in IPCC guidelines (Table 2.7 in IPCC, 2014). We suggest our new and more extensive field-measured values are more appropriate and that this involves significant adjustments for the EFs for most gases compared to the 2003 study notably: $CO_2$ (−8%), $CH_4$ (−55%), $NH_3$ (−86%) and CO (+39%). Improved EFs, at least for Kalimantan, for numerous other gases are found in Table 2. Finally, this work also

provides previously unavailable field measurements of aerosol optical properties. Both the aerosol and trace gas data in this study should be used with the understanding that many quantities will be affected by smoke evolution (e.g. Hobbs et al., 2003; Abel et al., 2003; Yokelson et al., 2009; Akagi et al., 2012; Alvarado et al., 2015).

### 3.5 Comparison to, assessment of, and synthesis with FLAME-4 lab data for peat fires

In this section we explore combining our new field data with the FLAME-4 lab data to develop an even more

comprehensive set of EFs for the peat component of peatland fires.

### 3.5.1. Lab/Field Comparison

Reasonable agreement for FLAME-4 lab measurements with our field measurements of aerosol properties was already demonstrated above. The comparison for the larger body of trace gas data is detailed next. For gases measured in FLAME-4 and the field for Kalimantan peat, and by Christian et al. (2003) in the lab for Sumatran peat,

we present the comparison graphically in Figure 7. Despite the high inherent variability, the Kalimantan field data overlap well with the Kalimantan samples burned in FLAME-4 (Stockwell et al., 2015). However, the one Sumatran peat sample is noticeably different. For the 21 compounds shown, 16 out of 21 field average EFs fall closer to Kalimantan lab mean EFs than the Sumatran lab EFs. However, based on one Sumatran sample alone we cannot yet say if the lab work is capable of resolving regional differences that may occur in peat fire emissions.

Table S3 compares all 31 gases measured for Kalimantan samples in both the lab (FLAME-4) and the field. The average of the two lab EFs is within a factor of two of the field mean for 20 of 31 species, which is adequate given that a factor of two is essentially also the field coefficient of variation ($n = 35$). In 7 of the 11 cases with more than a factor of two difference, the lab value is actually the sum of isomers compared to a single isomer from the GC analysis of the field WAS samples. For the remaining 4 species the lab values tend to be higher for unclear reasons.

For instance formic acid is higher in the lab where an open-path system was used instead of the closed cell system in



the field, which could be subject to sample losses. However, HCl and $NH_3$ are likely more prone to adsorption than formic acid (Yokelson et al., 2003) and they were higher as measured with the field system suggesting the Teflon sample line and coating on the closed cell were effective in limiting line losses. The lab average for NO is higher, but NO was below detection in one lab fire and high in the other where flaming briefly occurred. The one field fire

where flaming was briefly observed (Plume C, Table S2) had a higher EF for NO than the lab fire where it was detected. Thus, further comparisons with more lab fires will clearly be useful, but it appears the trace gas EFs from the lab are reasonable proxies for the EFs for species that have not been measured in the field.

### 3.5.2 Value of lab data

The value of lab NMOG EFs for peat burning is evident in at least two ways. First, with more powerful instruments

in the FLAME-4 study a significantly larger amount of NMOG mass was measureable. For the two FLAME-4 "stack" burns of Kalimantan peat (fires 114 and 125) where losses on the laboratory walls cannot occur during storage as with "room" burns (Stockwell et al. 2014), the high-resolution mass spectrometer and FTIR combined to measure 52.7 ± 5.0 g/kg total NMOG on average (Stockwell et al., 2015). This includes unidentified or tentatively assigned mass peaks that accounted for ~37% of detected NMOG mass. Our field equipment (with higher mobility

requirements) measured 22.5 ± 6.7 (max 30.3) g/kg of total NMOG emissions on average. An alternate metric is to note that the species measured in both the field and lab accounted for 52-68% of the total NMOG measured in the lab. In addition, a much larger number of species (>400) including extensive speciation of isomers by 2D-GC was reported in FLAME-4 although most of them were not emitted in large amounts (Hatch et al., 2015). Perhaps most importantly, the FLAME-4 lab experiment provides EFs for some key individual species not measured in the field

including: acrolein (an important air toxic, EF 0.19 ± 0.03 g/kg); methylgyloxal (important in the formation of both aqueous SOA and BrC (Lin et al., 2015), EF 0.19 ± 0.04 g/kg); and acetamide and other air toxics, which we discuss in more detail next.

The pure smoldering Kalimantan peat in FLAME-4 (fire 114) emitted acetamide (4.21 g/kg) at twice the mass of $NH_3$ (2.02 g/kg) (Stockwell et al., 2015). Acetamide can have numerous serious health effects (Ge et al., 2011) and

is considered a carcinogen by the International Agency for Research on Cancer (www.iarc.fr). Barnes et al. (2010) report that isocyanic acid (HNCO) and CO are the major oxidation products of acetamide, and small amounts of $CH_3OH$ and HCOOH formation are also seen. The acetamide lifetime would be ~3.3 days based on the measured OH rate constant ($0.35 \pm .1 \times 10^{11}$ $cm^3$ molecule$^{-1}$ s$^{-1}$) (Barnes et al., 2010). Acetamide also reacts quickly with Cl atoms, which could be important given Indonesia's common description as the "Maritime Continent." The main

oxidation product HNCO has a longer lifetime and is also of major concern for health effects as discussed by Roberts et al. (2011).

Akagi et al. (2014) discussed air toxic gases measured in biomass burning smoke in general terms and George et al. (2016) discussed hazardous air pollutants observed in lab measurements of burning coastal North Carolina peat. In Table 4 of Akagi et al. (2014), 26 air toxic gases in addition to CO that have been measured in smoke on a

reasonably frequent basis are shown along with recommended exposure limits. We measured 15 of these gases in the field (namely acetaldehyde, acetone, ammonia, benzene, 1,3-butadiene, ethylene, formaldehyde, HCl, *n*-hexane,





hydrogen cyanide (also a biomass burning tracer), methanol, phenol, styrene, toluene, and xylene). Six of the 9 others were measured for lab peat fires in FLAME-4 (acetonitrile [also a biomass burning tracer], acrolein, acrylonitrile, crotonaldehyde, methylethylketone, and naphthalene). Three of the 26 air toxics have markedly lower exposure limits than the others: formaldehyde, acrolein, and benzene. Our field-WAS EF benzene and lab

measurement by on-line mass spectrometry of EF benzene for smoldering Kalimantan peat burning agreed within 5 %. Our lab FTIR average EF HCHO is 77 % higher than our field FTIR average EF HCHO though 6 of the field fires had EF HCHO that were similar to or higher than the lab average ($n$=2).

Akagi et al. (2014) outline a method to estimate exposures using emission ratios that we can adapt here as a simple screening procedure for local exposure to air toxics in Kalimantan. We plan more detailed assessment of health

effects using the filter data (Jayarathne et al., in preparation, 2016) and regional $PM_{10}$ and visibility monitoring (Putra et al., in preparation, 2016). As mentioned earlier, regional $PM_{10}$ hit a maximum reported hourly reading of 3741 µg/m$^3$ in Palangkaraya, which, based on preliminary CO/PM ratios derived from Tables 1 and 2, would suggest a maximum hourly average of about 40 ppm CO (note, we did not monitor CO in Palangkaraya). This is similar to the recommended 8-hr limits (25-50 ppm) and well below the peak exposure limit of 200 ppm (Table 3,

Akagi et al., 2014). Using our HCHO/CO ratio from Table 1, the peak HCHO (ignoring chemical evolution) would be about 0.1 ppm. This is near the low end of various recommended peak exposure limits for HCHO indicating that HCHO exposure could be a concern for local residents. In addition, the synergistic health effects of multiple pollutants need more attention (Akagi et al., 2014).

## 4 Conclusions

During the strong 2015 El Niño event we deployed a mobile suite of ground-based trace gas and aerosol instruments in Central Kalimantan on the island of Borneo to make rare or unique field measurements of the fresh smoke emissions from fires burning peat of various types and at a range of depths. We report emission factors (EFs, g/kg) for the major greenhouse gases and about 90 gases in all obtained by Fourier transform infrared spectroscopy and whole air sampling. The EFs can be used with estimates of peat fuel consumption to improve regional emissions

inventories and assessments of the climate and health impacts of peatland fires. Our field data provide regionally-appropriate EFs for most of the measured gases that should be preferable to previously recommended EFs that were based on lab measurements of a single sample of smoldering Sumatran peat. Many of our new EF differ considerably from the previous recommendations; for example; $CO_2$ (–8%), $CH_4$ (–55%), $NH_3$ (–86%), CO (+39%), etc. The modified combustion efficiency of the peat fire smoke ranged from 0.693 to 0.835 with an average of 0.772

± 0.035 ($n$=35) indicating essentially pure smoldering combustion and no significant lofting of the initial emissions was observed. EFs (g/kg) for major gas-phase tracers, air toxics, or carcinogens measured include: HCN (5.8 ± 1.6), formaldehyde (0.87 ± 0.48), BTEX (benzene, toluene, ethylbenzene, xylenes, 1.5 ± 0.6), and 1,3-butadiene (0.19 ± 0.16). The field results from Kalimantan were in reasonable agreement with recent (FLAME-4) lab measurements of the trace gases and aerosol from smoldering Kalimantan peat for species measured in both studies. This suggests lab

measurements can provide useful EFs for species not yet measured in the field such as the air toxics acrolein (0.19 ± 0.03 g/kg), and acetamide (2.54 ± 2.36 g/kg). Except for HCN (lifetime in months) and benzene (lifetime in days),





these air toxics observed in the field and FLAME-4 are all reactive and, therefore, of most concern for local exposure. A simple screening procedure suggests that formaldehyde and the synergistic effects of multiple pollutants are most likely to challenge recommended exposure limits locally. HNCO as a longer-lived photochemical product of acetamide could be a health concern regionally.

In addition, we measured in-situ aerosol optical properties at 405 and 870 nm with two photoacoustic extinctiometers and analyzed particulate collected on filters. The aerosol optical data measured include EFs for the scattering and absorption coefficients (EF $B_{scat}$ and EF $B_{abs}$, $m^2$/kg fuel burned) and SSA at both wavelengths. Consistent with the minimal flaming combustion, the emissions of BC were negligible (0.0055 ± 0.0016 g/kg) and aerosol absorption was overwhelmingly due to the organic component. For example, brown carbon contributed

~96% of aerosol absorption at 405 nm and absorption at 405 nm was ~52 times larger than at 870 nm. The importance of the organic absorption was also seen in the high average AAE (4.97 ± 0.65, range 4.29-6.23) and the average SSA at 405 nm (0.974 ± 0.016) being lower than the average SSA at 870 nm (0.998 ± 0.001). However, comparing the $B_{abs}$ at 405 nm to the simultaneously measured organic carbon mass on filters suggests a low MAC (~0.1 $m^2$/g) for the bulk OC, as expected for the low BC/OC ratio in the aerosol.

Future lab measurements of burning peat should be useful to screen for regional differences in emissions based on geographic origin, distance from the coast, etc., and to extend the measurement capability to new gases (e.g. highly oxygenated NMOG) and aerosol properties (e.g. size distribution, cloud condensation nuclei activity, OA volatility, etc.). Ground-based measurements of peat fire emissions in other regions of Southeast Asia are needed. In addition, an extensive regional airborne campaign is critically needed for characterization of the mix of fire types that

currently dominate the overall region and to measure the detailed evolution of the peatland fire smoke plumes and the coalesced regional haze.

**Acknowledgements.** This research was primarily supported by NASA Grant NNX13AP46G to SDSU and UM. The research was also supported by NASA grant NNX14AP45G to UM. Purchase and preparation of the PAXs was

supported by NSF grant AGS-1349976 to R. Y. We thank G. McMeeking, J. Walker, and S. Murphy for helpful discussions on the PAX instruments and data analysis. This work would not have been possible without the excellent support provided by the BOS office in Palangkaraya; notably Laura Graham, Grahame Applegate, and the BOS field team.

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



**Table 1.** Study-average emission factors (g/kg) and one standard deviation (stdev) for trace gases significantly elevated above background in Kalimantan peat fire plumes.

| Compound (formula) | Study avg (stdev) 35 plumes |
|---|---|
| MCE | 0.772(0.035) |
| Carbon Dioxide ($CO_2$) | 1564(77) |
| Carbon Monoxide (CO) | 291(49) |
| Methane ($CH_4$) | 9.51(4.74) |
| Dihydrogen ($H_2$) | 1.22(1.01) |
| Acetylene ($C_2H_2$) | 0.121(0.066) |
| Ethylene ($C_2H_4$) | 0.961(0.528) |
| Propylene ($C_3H_6$) | 1.07(0.53) |
| Formaldehyde (HCHO) | 0.867(0.479) |
| Methanol ($CH_3OH$) | 2.14(1.22) |
| Formic Acid (HCOOH) | 0.180(0.085) |
| Acetic Acid ($CH_3COOH$) | 3.89(1.65) |
| Glycolaldehyde ($C_2H_4O_2$) | 0.108(0.089) |
| Furan ($C_4H_4O$) | 0.736(0.392) |
| Hydroxyacetone ($C_3H_6O_2$) | 0.860(0.433) |
| Phenol ($C_6H_5OH$) | 0.419(0.226) |
| 1,3-Butadiene ($C_4H_6$) | 0.189(0.157) |
| Isoprene ($C_5H_8$) | 5.28E-2(4.33E-2) |
| Ammonia ($NH_3$) | 2.86(1.00) |
| Hydrogen Cyanide (HCN) | 5.75(1.60) |
| Nitrous Acid (HONO) | 0.208(0.059) |
| Hydrogen chloride (HCl) | 3.46E-2(2.05E-2) |
| Nitric Oxide (NO) | 0.307(0.360) |
| Carbonyl sulfide (OCS) | 0.110(0.036) |
| DMS ($C_2H_6S$) | 2.82E-3(2.34E-3) |
| Chloromethane ($CH_3Cl$) | 0.147(0.057) |
| Bromomethane ($CH_3Br$) | 1.01E-2(3.52E-3) |
| Methyl iodide ($CH_3I$) | 1.25E-2(4.48E-3) |
| Dibromomethane ($CH_2Br_2$) | 1.04E-4(7.70E-5) |
| Ethane ($C_2H_6$) | 1.52(0.66) |
| Propane ($C_3H_8$) | 0.989(0.644) |
| i-Butane ($C_4H_{10}$) | 9.11E-2(1.02E-1) |
| n-Butane ($C_4H_{10}$) | 0.321(0.225) |
| 1-Butene ($C_4H_8$) | 0.182(0.085) |
| i-Butene ($C_4H_8$) | 0.311(0.160) |
| trans-2-Butene ($C_4H_8$) | 7.75E-2(3.80E-2) |
| cis-2-Butene ($C_4H_8$) | 6.15E-2(3.34E-2) |
| i-Pentane ($C_5H_{12}$) | 0.123(0.135) |
| n-Pentane ($C_5H_{12}$) | 0.243(0.131) |
| 1,2-Propadiene ($C_3H_4$) | 1.84E-3(2.27E-3) |
| Propyne ($C_3H_4$) | 5.65E-3(8.57E-3) |
| 1-Butyne ($C_4H_6$) | 1.98E-3(1.37E-3) |
| 2-Butyne ($C_4H_6$) | 1.15E-3(1.51E-3) |
| 1,3-Butadyne ($C_4H_2$) | 2.99E-4(2.42E-4) |
| 1,2-Butadiene ($C_4H_6$) | 6.15E-4(6.39E-4) |
| 1-Pentene ($C_5H_{10}$) | 0.110(0.066) |
| trans-2-Pentene ($C_5H_{10}$) | 3.97E-2(2.76E-2) |
| cis-2-Pentene ($C_5H_{10}$) | 2.24E-2(1.52E-2) |
| 3-Methyl-1-butene ($C_5H_{10}$) | 3.03E-2(1.98E-2) |
| 2-Methyl-1-butene ($C_5H_{10}$) | 2.99E-2(1.61E-2) |
| 2-Methyl-2-butene ($C_5H_{10}$) | 6.47E-2(3.72E-2) |
| 2-Methyl-1-Pentene ($C_6H_{12}$) | 0.109(0.076) |
| 1,3-Pentadiene ($C_5H_8$) | 1.98E-2(1.04E-2) |
| 1,3-Cyclopentadiene ($C_5H_6$) | 9.98E-3(5.85E-3) |
| Cyclopentene ($C_5H_8$) | 2.46E-2(1.57E-2) |
| 1-Heptene ($C_7H_{14}$) | 7.90E-2(5.40E-2) |





| | |
|---|---|
| 1-Octene ($C_8H_{16}$) | 6.52E-2(4.24E-2) |
| 1-Decene ($C_{10}H_{20}$) | 4.98E-2(3.88E-2) |
| n-Hexane ($C_6H_{14}$) | 0.143(0.087) |
| n-Heptane ($C_7H_{16}$) | 0.112(0.074) |
| n-Octane ($C_8H_{18}$) | 9.80E-2(6.90E-2) |
| n-Nonane ($C_9H_{20}$) | 8.95E-2(6.33E-2) |
| n-Decane ($C_{10}H_{22}$) | 7.44E-2(5.09E-2) |
| 2,3-Dimethylbutane ($C_6H_{14}$) | 5.31E-3(4.15E-3) |
| 2-Methylpentane ($C_6H_{14}$) | 3.97E-2(3.58E-2) |
| 3-Methylpentane ($C_6H_{14}$) | 9.31E-3(8.00E-3) |
| Benzene ($C_6H_6$) | 0.954(0.394) |
| Toluene ($C_7H_8$) | 0.370(0.306) |
| Ethylbenzene ($C_8H_{10}$) | 4.17E-2(2.02E-2) |
| m/p-Xylene ($C_8H_{10}$) | 0.122(0.055) |
| o-Xylene ($C_8H_{10}$) | 0.103(0.059) |
| Styrene ($C_8H_8$) | 2.71E-2(1.31E-2) |
| i-Propylbenzene ($C_9H_{12}$) | 5.34E-3(3.74E-3) |
| n-Propylbenzene ($C_9H_{12}$) | 1.18E-2(8.20E-3) |
| 3-Ethyltoluene ($C_9H_{12}$) | 2.70E-2(2.28E-2) |
| 4-Ethyltoluene ($C_9H_{12}$) | 2.35E-2(2.13E-2) |
| 2-Ethyltoluene ($C_9H_{12}$) | 4.16E-2(3.35E-2) |
| 1,3,5-Trimethylbenzene ($C_9H_{12}$) | 1.08E-2(8.55E-3) |
| 1,2,4-Trimethylbenzene ($C_9H_{12}$) | 6.96E-2(5.52E-2) |
| 1,2,3-Trimethylbenzene ($C_9H_{12}$) | 6.39E-2(4.57E-2) |
| alpha-Pinene ($C_{10}H_{16}$) | 2.99E-3(2.88E-3) |
| beta-Pinene ($C_{10}H_{16}$) | 1.67E-3(1.76E-3) |
| 2-Methylfuran ($C_5H_6O$) | 0.121(0.123) |
| Nitromethane ($CH_3NO_2$) | 6.01E-2(3.10E-2) |
| Acetaldehyde ($C_2H_4O$) | 0.697(0.460) |
| Butanal ($C_4H_8O$) | 2.38E-2(1.91E-2) |
| Furfural ($C_5H_4O_2$) | 0.124(0.116) |
| Acetone ($C_3H_6O$) | 0.691(0.356) |
| Butanone ($C_4H_8O$) | 0.136(0.068) |
| Methyl vinyl ketone ($C_4H_6O$) | 5.69E-2(4.27E-2) |





Table 2. Aerosol emission factors and optical properties measured by the PAX and filter sampling.

| Plume ID> | Q | R[a] | S | T[a] | V | W[a] | W[a] | PAX (7) avg (stdev) |
|---|---|---|---|---|---|---|---|---|
| Date> | 5-Nov | 5-Nov | 5-Nov | 5-Nov | 6-Nov | 6-Nov | 6-Nov | |
| Filter # | 21 | 22 | 23 | 24 | 25 | 27 | 28 | |
| EF BC (g/kg) | 5.23E-3 | 5.49E-3 | 5.27E-3 | 6.62E-3 | 8.32E-3 | 4.45E-3 | 3.22E-3 | 5.52E-3(1.62E-3) |
| EF $B_{abs}$ 870 (m$^2$/kg) | 2.48E-2 | 2.60E-2 | 2.50E-2 | 3.14E-2 | 3.95E-2 | 2.11E-2 | 1.53E-2 | 2.61E-2(7.66E-3) |
| EF $B_{scat}$ 870 (m$^2$/kg) | 7.84 | 26.9 | 19.3 | 21.2 | 21.4 | 17.9 | 13.5 | 18.3(6.1) |
| EF $B_{abs}$ 405 (m$^2$/kg) | 2.91 | 1.33 | 0.787 | 1.61 | 1.78 | 0.651 | 0.405 | 1.35(0.85) |
| EF $B_{scat}$ 405 (m$^2$/kg) | 46.2 | 60.9 | 37.3 | 78.6 | 52.7 | 43.6 | 34.9 | 50.6(15.2) |
| EF $B_{abs}$ 405 just BrC (m$^2$/kg) | 2.85 | 1.29 | 0.733 | 1.54 | 1.69 | 0.606 | 0.374 | 1.30(0.85) |
| EF $B_{abs}$ 405 just BC (m$^2$/kg) | 5.32E-2 | 4.22E-2 | 5.36E-2 | 6.74E-2 | 8.48E-2 | 4.54E-2 | 3.13E-2 | 5.40E-2(1.76E-2) |
| SSA 870 nm | 0.997 | 0.999 | 0.999 | 0.999 | 0.998 | 0.999 | 0.999 | 0.998(0.001) |
| SSA 405 nm | 0.941 | 0.979 | 0.979 | 0.980 | 0.967 | 0.985 | 0.989 | 0.974(0.016) |
| AAE | 6.23 | 5.14 | 4.51 | 5.15 | 4.98 | 4.49 | 4.29 | 4.97(0.65) |
| MCE real-time | 0.726 | 0.763 | 0.773 | 0.778 | 0.824 | 0.833 | 0.831 | 0.790(0.041) |
| MCE grab sample | 0.693 | 0.761 | 0.779 | 0.795 | 0.824 | 0.835 | 0.835 | 0.789(0.051) |
| EF PM$_{2.5}$ (g/kg)[b] | 19.3 | 21.5 | 17.9 | 29.6 | 24.3 | 22.5 | 15.7 | 21.5(4.6) |
| EF OC (g/kg)[b] | 10.5 | 16.7 | 13.6 | 26.9 | 14.9 | 17.6 | 11.6 | 16.0(5.5) |
| EF EC (g/kg)[b] | 0.386 | 0.175 | 0.196 | 0.258 | 0.354 | 0.237 | 8.98E-02 | 0.242(0.103) |
| MAC est. (405) (m$^2$/g) | 0.271 | 7.69E-2 | 5.40E-2 | 5.71E-2 | 1.14E-1 | 3.45E-2 | 3.22E-2 | 9.13E-2(8.38E-2) |

a-For these plumes, PAX and filter collection times are completely in sync.
b-For these quantities an average based on all the filter samples will be reported by Jayarathne et al., (2016 in prep).




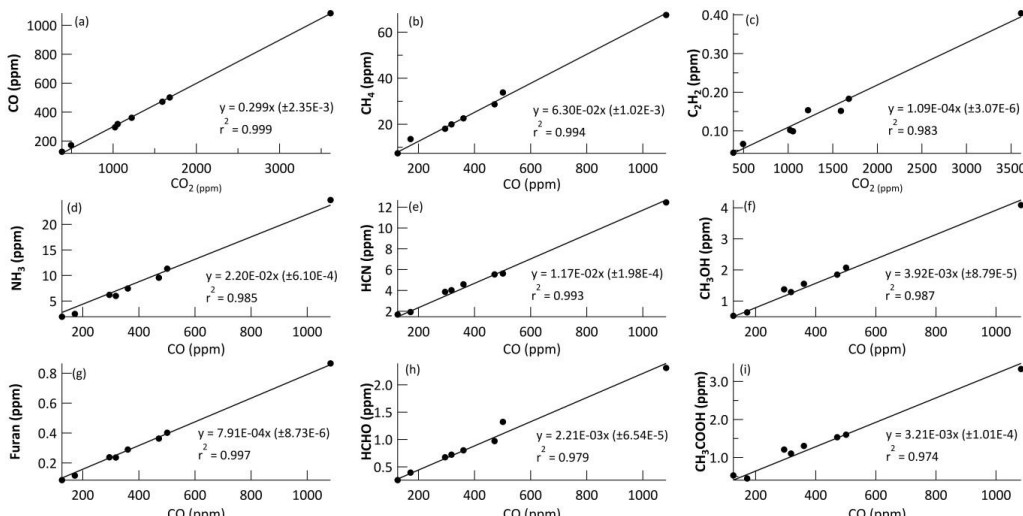

**Figure 1.** ER plots from plume N for (a) carbon monoxide, (b) methane, (c) acetylene, (d) ammonia, (e) HCN, (f) methanol, (g) furan, (h) formaldehyde, and (i) acetic acid measured by FTIR.





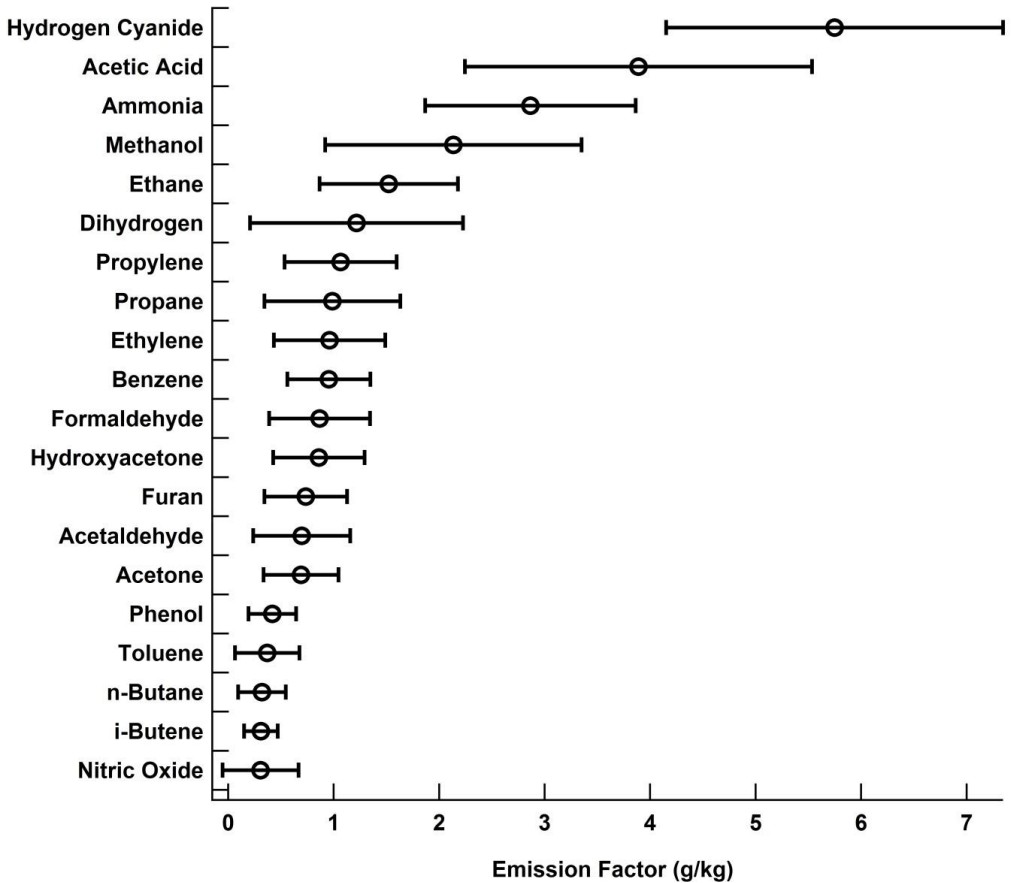

**Figure 2.** The emission factors (g/kg) and ± one standard deviation for the 20 most abundant trace gases (excluding $CO_2$, CO, $CH_4$) in this dataset.

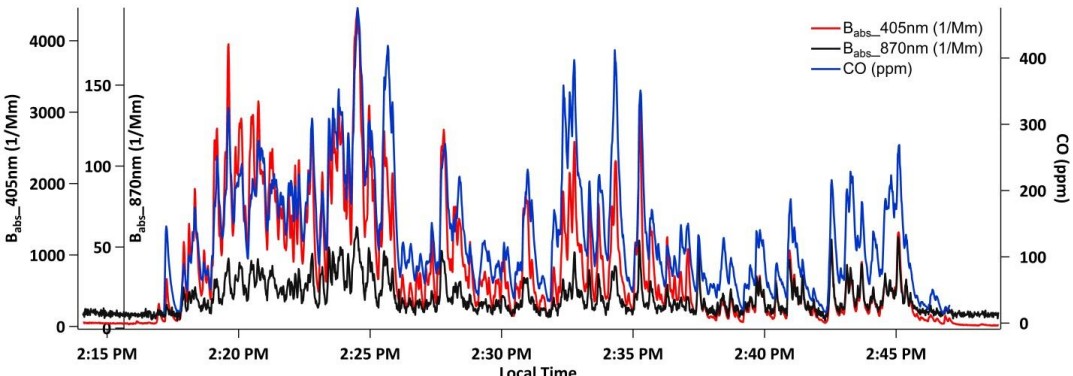

**Figure 3.** PAX real-time $B_{abs}$ at 870 (black) and 405 (red) nm collected on 5 November showing the dominance of absorbing aerosol at 405 nm. The co-located CO mixing ratio measurement from the real-time FTIR data is shown in blue. CO background was obtained from grab samples for increased accuracy. A transition to more glowing dominated combustion with a lower aerosol to CO ratio (and lower AAE and higher MCE, not shown) is observed at about 2:37 pm.





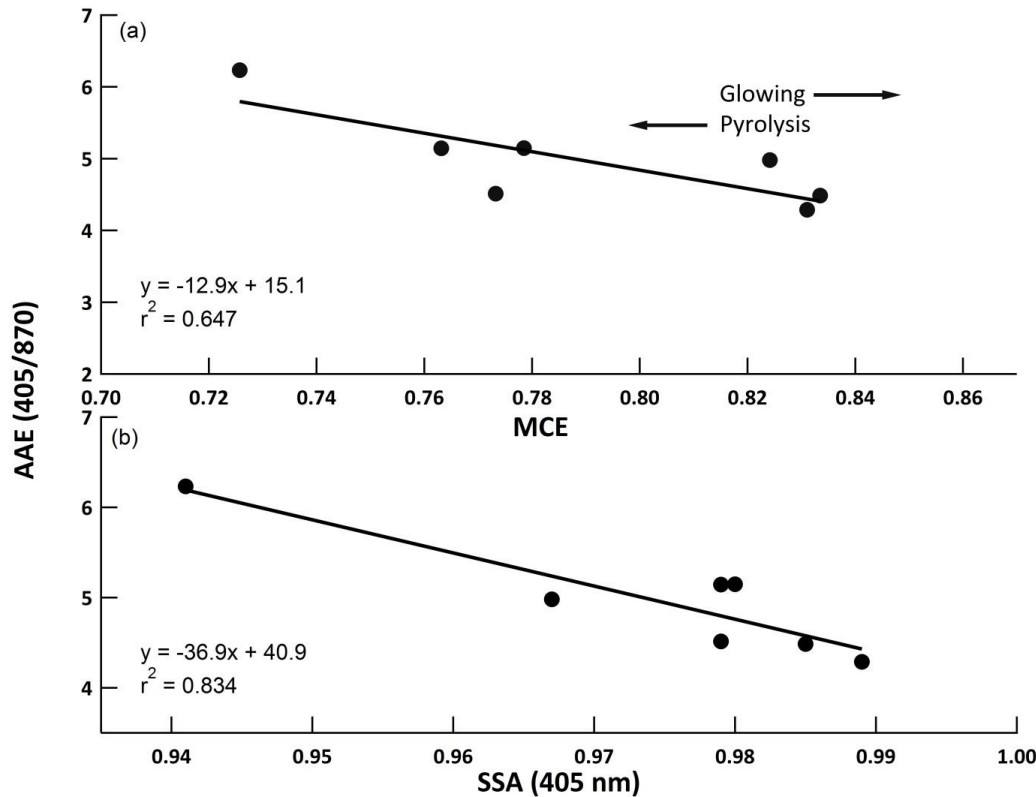

**Figure 4.** Correlations of (a) AAE versus MCE and (b) AAE vs SSA (405 nm).





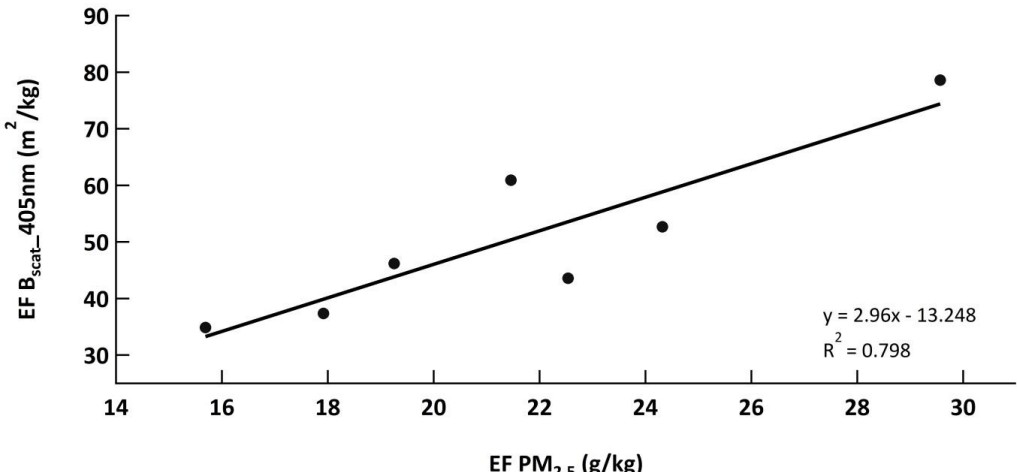

**Figure 5.** The emission factor of $B_{scat}$ at 405 nm versus $PM_{2.5}$ EF. The slope is an estimate of the mass scattering efficiency.




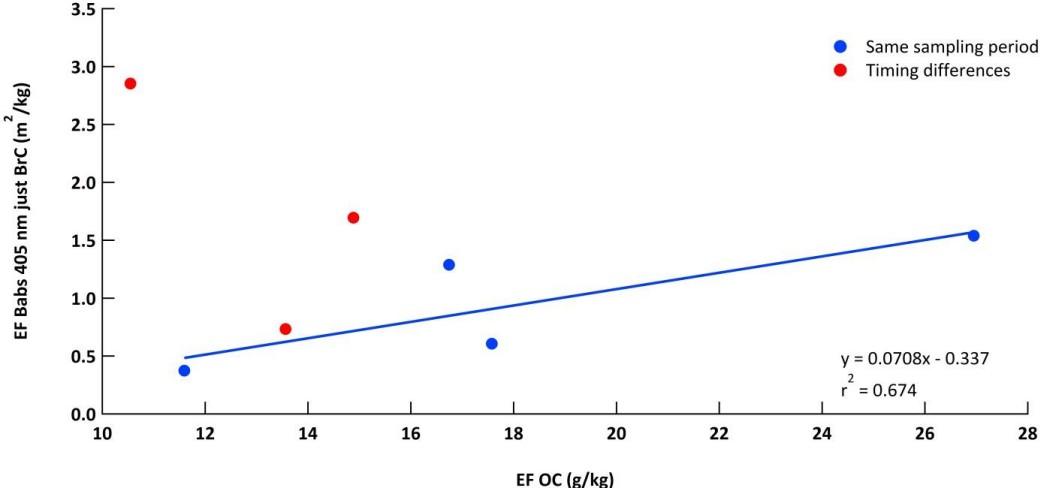

**Figure 6.** The estimated mass absorption coefficient of the bulk OC from the $B_{abs}$ assigned to BrC versus simultaneously measured OC mass on filters. Only 4 plumes were sampled by both techniques over the exact same time period (blue symbols) and they were used in fit shown to estimate the MAC.





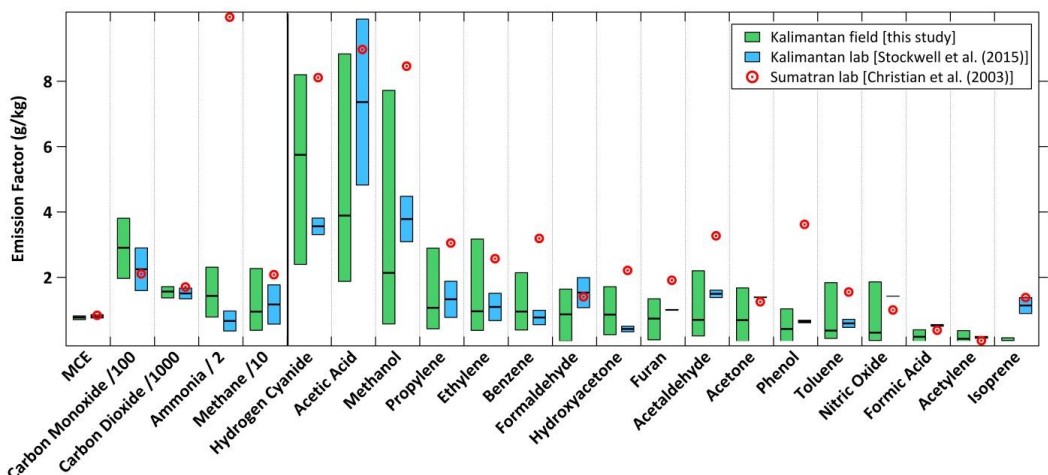

**Figure 7.** Study overlap (minimum, maximum, and average) including field Kalimantan samples from this study (green), Kalimantan laboratory stack burns (blue; Stockwell et al., 2014; 2015) and a single laboratory burn of Sumatran peat (red; Christian et al., 2003).