# Peer review of "Field measurements of trace gases and aerosols emitted by peat fires in Central Kalimantan, Indonesia during the 2015 El Niño"

_Atmospheric Chemistry and Physics, 2016_

## Referee Comment (RC1) · Anonymous Referee #2 · 16 Jul 2016

Review of "Field measurements of trace gases and aerosols emitted by peat fires in Central Kalimantan, Indonesia during the 2015 El Niño", by C. E. Stockwell et al., 2016.

The manuscript by Stockwell et al. presents measurements from 2015 peatland fires in the Indonesian province Kalimantan. The findings presented in this manuscript both add to and modify previous lab-based measurements of peat combustion, amending a handful of key EFs that were previously only available from laboratory studies, while confirming the validity of laboratory studies for estimating EFs for species that are not easily measured in the field. The paper is well-written, cohesive and thorough, and my assessment is that it merits publication in ACP after the following issues are addressed.

Specific Comments

[Figure]

Page 2, line 19 – "(2012)" - is that a reference? It should be included properly here.

Page 4, line 32 and Page 19, line 10 – I don't think it's necessary to put "in preparation" here – it is included in the reference itself. However, please include a full reference for this work if possible, including a full author list and title.

Page 5, lines 11-12 – please explain what n=1 is in reference to, or simply state, if this is the case, that there was only a single sample of each type analyzed in the lab study.

Page 5, line 33 – it would be helpful to the reader to direct them to Table S1 at this point, rather than making them wait to find out about the table until the next page.

Page 6, lines 4-25 – a map/diagram of the sites would help put this entire sampling description into context.

Page 12, lines 20-25 – where are the ERs if I want to look at them? A lot of this discussion is very qualitative and vague (i.e. "seven of these nine cases agree..." – what about the other two? And what were they? It would seem reasonable to spell that out or offer something specific about the differences between the FTIR and the WAS sample that would alleviate the reader's concern that there is something we should know about the alleged differences. I understand that the analysis were not set up to evaluate differences, and yet to just allude to it but not give us anything further is more suspicious.

Page 12, lines 33-35 – many of the uncertainties are unreasonably precise – i.e., 0.867 $\pm$ 0.479 and 0.860 $\pm$ 0.433. Please round these to make them more reasonable for reporting. Page 13, lines 2-3 the "overlap" isn't very surprising, considering that the range in your work is from 0.3 to 1.44. Overlap isn't hard, and likely shouldn't be emphasized like this. "Are consistent", perhaps.

Page 13, lines 10-15 – I'm not fond of the idea of alluding to something that should be done, and then just saying "we haven't attempted this yet." Why bring it up? Or why not attempt to include the analysis here?

Page 14, lines 5-16 – I have issues with this plot, and with the implication that the overlap in time is so fortuitously going to take something with 7 points, eliminate 3, and leave you with a four point plot that has an r^2 of 0.674, and that you're going to give it any actual credence. I don't think you "confirmed" the MAC near 0.1 at all. You just eliminated points until the remainder of your points came slightly close to giving you a line. This either needs far more justification, or it shouldn't be included.

Page 14, line 36 – please comment on the differences between the Liu et al. paper using SSA 781 and you using SSA 870 nm, and what kind of linearity you expect for the two different locations, and how that affects your comparison. Also, this comparison of the observed aerosol parameters with literature values would benefit from having a table like S3 included.

Page 15, line 36 – "and other factors" is very vague. Please expound.

Page 16, line 13 – you "interpolated" between two points to find something. How did you do this? Was it linear? Why? How do you know?

Page 15-16, Section 3.3. This section feels very hand-wavy. I would like to see a more quantitative and step-wise analysis presented for the comparisons mentioned to previous studies and other kinds of peat BB observations in this paragraph. Some of the comparisons mentioned are presented with little defense as to their relevance and/or the validity of the comparison (i.e., the interpolation mentioned above.)

Page 17, line 33 – "the lab value is actually the sum of isomers compared to a single isomer from the GC analysis..." please explain this more, including references to the table, in which I see no evidence of a difference between a sum of isomers and a single isomer. Is this for a particular compound or set of compounds? Be specific.

Page 18, line 14 – you should reassure the reader that the ∼37% of unidentified or tentatively assigned mass peaks of the NMOG mass is not going to negatively affect your assumption that you are measuring all the carbon to be factored into your EF

calculations. I'm sure it's not significant, considering the major non-NMOG carbon species, but this should be recognized.

In Table 1, Table 2, Figure 1, etc., there are numbers that are both too precise considering the standard deviations reported, and I dislike the excel-style presentation of numbers with exponents written as (e.g.) 1.67E-3.

Re: Table S1 – there are a handful of things that would make this table easier to digest, without having to search out other information. Instead of "Y/N type" or "Y/N what" as a header, eliminate Y/N and just include the type/what or put "none" or "unknown" where applicable. Also, please spell out here what the peat fuel types are, so that I don't have to go back and find that in the paper (in the footnote would be fine.) "day-mon" should be "DD-Mon". Why are there some plumes included that aren't lettered? These don't seem to add anything to the paper. "seec" is not a word or a shortform (as a direction). For Depth of Burn, " $\sim$ site avg" is redundant. "site avg" is fine. Why is so little known about site 6? Re: Winds – "av, max, dir" implies you'll have numeric values below. Maybe leave the "av, max, dir" part out, and just consider it a verbal description of the winds. Be consistent with spacing and vertical cell centering. Also, for all three supporting information tables (S1-S3), please be consistent about font size and styles and remove bold settings. Table titles should all be uniformly sized. For all supplement tables, if these are being submitted as they are now in an excel file, a san serif font is likely best for readability. If you're preparing a printed document, a simple serif font is also acceptable (i.e., Times New Roman.)

Technical Comments Page 5, lines 4-5 – don't us semicolons in place of commas. Page 5, line 26 – remove the hyphen from "at six-different peatland..." Page 6, line 10 – no need for "(#2)" after "This site..." Page 11, line 12 – there is a missing or extra parenthesis here. Page 18, line 28 – "0.35 $\pm$ .1 x 10..." – the .1 should be 0.10. Page 19, line 1 – "Six of the nine..." Page 19, line 11 – the Putra et al. paper in preparation needs to be included in the reference list. Tables 1, S2 and S3: "ethyne", "ethene", "propene" (and in the text and Figure 1, where applicable).

---

## Referee Comment (RC2) · Anonymous Referee #1 · 18 Jul 2016

The manuscript "Field measurements of trace gases and aerosols emitted by peat fires in Central Kalimantan, Indonesia during the 2015 El Niño" presents the first field emission measurements of comprehensive atmospheric compositions from peat fires burning in Southeast Asia. This kind of field measurements is extremely rare and thus very valuable to the scientific community of Atmospheric Chemistry and Physics. The measurement methods used in the study are well established and the field experiment design is reasonable and justified. I expected this manuscript would only reported emissions from a rarely studied environment (which itself would add values to literature), but the discussion on the representativeness of the field measurements, comparison to previously available emission factors for the same type of emissions

is very useful too. The authors also compare the field measured emission factors to those obtained from lab experiments, and discuss the value and importance of lab data. Peat fire burning in Southeast Asia is such an interesting and important topic from atmospheric chemistry and climate perspectives but many questions still remain as first order research problems due to the limited field data. I believe the manuscript could be much improved in terms of how to scale the field data to a large spatial area in this region, but I understand that the study is also limited by prior data and resource that could be deployed. The manuscript is well written in general, while the readability could be improved by properly introducing acronyms. In summary, I think this manuscript could be published and I list a few minor suggestions as below:

1. The manuscript points the importance and uniqueness of 2015 El Niño event. The authors need to comment on how this field measurements during a El Niño event apply to other 'normal' years, or do the authors suggest that these field measured emission factors can only apply to El Niño events? Can the difference between lab and field comparison be partly explained by the special El Niño event?

2. Related to point 1: This manuscript finds that many significant revisions of emission factors compared previously widely used EFs, mostly reductions ($CO_2$, $CH_4$, $NH_3$). But as the authors point in the introduction, previous studies suggest "in Southeast Asia, in the 1980s-1990s, peatland fires were a major source of carbon to the atmosphere mainly during El Niño induced droughts . . .. " How can the authors reconcile this? The manuscript uses "the 2015 El Niño" in the title, and the authors would be expected to comment more on this event. However, such comments are very rare in this version of the manuscript.

3. P5 L14: the definition of fuel moisture is not clear. What is 'wet', what is 'dry'? Here and many places in the manuscript, the authors assume all readers know most of acronyms related to fire studies. Properly introducing them could help the manuscript reach a broad audience of atmospheric scientists.

4. P7 L29: 'cyclones' should be 'cyclone samplers'? The authors need to avoid using 'field language' as much as possible and try to use its formal name.

5. P8 L15: poorly written.

6. P8 L21-22: here and other places, the instrument modes and manufactures should be listed as full names with company names and locations.

7. P 9, session 2.2.5: it is unclear if any control (or blank) samples were deployed for these offline measurements? For example, pre-cleaned filters shipped with other filters but without any sampling.

8. P13 L13-15: it would be very valuable if those peat characteristics were mapped and it could help to scale these point measurements to large areas and perhaps devise a parameterization to study other peat fire emission. It is very unfortunate that this study did not attempt such an analysis. What would be the authors' recommendations to future field studies? It would be useful for other researchers who are interested in this area.

9. P15 L 36: what is 'rapid green-up'. Again, the manuscript could be improved and reach a broader audience if these words were properly defined.

---

## Referee Comment (RC3) · Anonymous Referee #3 · 29 Jul 2016

This paper presents results of a field campaign measuring emissions from burning Indonesian peat in-situ, during the intense burning during the intense 2015 El Nino event. These very challenging measurements were collected with mobile sampling set up incorporating FTIR to measure a range of gas phase species, several photoacoustic extinctiometers to measure aerosol light absorption and scattering at two wavelengths and filter and canister samplers to collected integrated samples of condensed- and gas-phase species, respectively. 35 separate plumes were measured with different combinations of instruments, resulting in measurements constraining both the central tendency and the (relatively large) variability in emission factors resulting from combustion of this fuel.

[Figure]

As evidenced by the startlingly high PM concentrations ( >3000 ug/m3) observed in nearby cities, peat combustion can and does make an enormous contribution to loadings of atmospheric aerosols and a wide range of gas-phases species. Considering that the only extant emission factors come from a handful of laboratory burns, which a) do not capture the potential variability in emissions and, ,b) may not recreate combustion conditions observed when the fuel is in place, there is great value in the results of this study. The measurements of gas and aerosol species and aerosol optical properties appear to be carefully conducted and are well documented in this manuscript, and I particularly applaud attention to uncertainty in measurements and the resulting propagated uncertainty in EFs (though echo the other referees' comments concerning their presentation in tables). A rather extensive effort is made to compare the results with those measured in earlier lab measurements, showing general consistency but some very significant differences. The resulting emission factors will be of great use for emission inventory development and chemical transport modeling to understand the impacts of the dramatic land use transformation taking place in this region. Therefore, the manuscript is certainly suitable for publication in ACP. Below I list several points of clarification that would enhance the readability and utility of the manuscript.

One general comment is that there is a relatively large number of portions of the text that seem overly detailed and make the paper harder to read than it might otherwise be. Ideally, these could be moved to an online supplemental section. While I understand the use of a spreadsheet for the supplement in this case, perhaps a second file could be used or some of these details moved to aid readers. Examples include: P. 6, L1-25 (description of sampling sites), P. 8, L22-28 (description of PAX operating principle), P. 10, L20-28 (description of alternate data reduction approaches).

Minor Points:

P9, L23-24 – This description is not clear and imprecise; for example, what is the standard deviation of smoke PM? I think I understand this to be the standard deviation of PM mass concentration, but since this is from a single filter, how is a standard

deviation determined? And why is 10% of PM concentration used?

P12, L3-4 – this is imprecise, BC does not absorb light 'proportional to frequency'.

P12, L25 – Unclear. What is meant by 'less close', and what do you mean 'not well-designed for comparison'?

P13, L11-12- While I get what you're saying, this is also imprecise. It doesn't necessarily follow that fires with both smoldering and flaming will have a linkage between MCE and EFs. I think what you mean is that in cases where you have a wider range of MCEs you tend to see a (anti) correlation between MCE and EFs, but the small range of low MCEs observed here means you don't see such a trend.

P13, L13-14 – Awkward sentence, what would such characteristics be?

P13, L18-19 – This evolution in the absorption in this plume seems to be also linked with CO emissions, with Bap(405nm) and CO very tightly correlated in the early stage of the burn, and much less so later on. Were stages of combustion typically seen? Is there any somewhat consistent trajectory in emissions during a burn? I assume that because you measured a relatively large number of plumes that you captured emissions from a range of stages, but it would be interesting to learn of any consistency to give some insights into how laboratory tests can be more representative of combustion observed in large fuel beds in-situ.

P14, L3-4 – 'obtained closer to 500 nm' is unclear, presumably this refers to illumination wavelength? Would be best to make these MSE values more directly comparable if at all possible.

P15, L16-18 – This sentence is very hard to read/understand. Overestimated by what, relative to what?

P15, L33 – Not sure what is meant by 'significant areas' or 'deep burn depth'. Please clarify.

[Figure]

P15, L35 – What is distinctly wrong about 'fire products used in top-down approaches'? This sentence is unclear.

P16, L1 – Following from previous confusing sentences, not sure what is meant by 'tend to cancel'.

P19, L4-7 – This is a bit of a non-sequitur as you are talking about exposure and health impacts, and then shift to an EF comparison concerning lab/field measurements which doesn't really so much apply to these very 'high level' estimates. If anything a consistency in air toxic/PM ratio could be highlighted, as this is what you're using to estimate air toxic exposure concentrations.

---

## Author Comment (AC2) · 6 Sep 2016

Response to Referee #2

We thank the Referee for their encouraging assessment and constructive suggestions, which will improve the paper. The Referee comments are reproduced below followed by our detailed response.

Anonymous Referee #2

Review of "Field measurements of trace gases and aerosols emitted by peat fires in Central Kalimantan, Indonesia during the 2015 El Niño", by C. E. Stockwell et al., 2016. The manuscript by Stockwell et al. presents measurements from 2015 peatland fires

in the Indonesian province Kalimantan. The findings presented in this manuscript both add to and modify previous lab-based measurements of peat combustion, amending a handful of key EFs that were previously only available from laboratory studies, while confirming the validity of laboratory studies for estimating EFs for species that are not easily measured in the field. The paper is well-written, cohesive and thorough, and my assessment is that it merits publication in ACP after the following issues are addressed.

Specific Comments

**R2.1:** Page 2, line 19 – "(2012)" - is that a reference? It should be included properly here.

**Authors:** We found "(2012)" on P2, L8. The year of the recent measurements was included because our field data did not agree well with the lab measurements made in 2001 that IPCC uses, but did agree well with lab measurements made in 2012. It's probably safe to delete the year and we have done so.

**R2.2:** Page 4, line 32 and Page 19, line 10 – I don't think it's necessary to put "in preparation" here – it is included in the reference itself. However, please include a full reference for this work if possible, including a full author list and title.

**Authors:** We removed "in preparation" throughout and updated the reference as suggested.

**R2.3:** Page 5, lines 11-12 – please explain what n=1 is in reference to, or simply state, if this is the case, that there was only a single sample of each type analyzed in the lab study.

**Authors:** Within both parentheses we changed "n=1" to "one sample"

**R2.4:** Page 5, line 33 – it would be helpful to the reader to direct them to Table S1 at this point, rather than making them wait to find out about the table until the next page.

**Authors:** Good suggestion and we now call out Table S1 earlier on P5, L27.

**R2.5:** Page 6, lines 4-25 – a map/diagram of the sites would help put this entire sampling description into context.

**Authors:** A site map has now been provided as Supplementary Figure 2 (now referenced on P6 L4).

**R2.6:** Page 12, lines 20-25 – where are the ERs if I want to look at them? A lot of this discussion is very qualitative and vague (i.e. "seven of these nine cases agree: : :" – what about the other two? And what were they? It would seem reasonable to spell that out or offer something specific about the differences between the FTIR and the WAS sample that would alleviate the reader's concern that there is something we should know about the alleged differences. I understand that the analysis were not set up to evaluate differences, and yet to just allude to it but not give us anything further is more suspicious.

**Authors:** This is an excellent comment and we considered adding this comparison to the supplement in response. For background, we had two WAS cans that the field notes indicated were filled from the FTIR cell. While changes could theoretically occur to some species during storage in the FTIR cell, this seemed like a valid opportunity to compare the data for the overlap species fairly directly. Unfortunately, in attempting to further clarify this comparison, we have now realized that the FTIR cell was actually re-filled after the FTIR measurements and before the cans were filled. Thus, the comparison included an unknown contribution from natural plume variability and was not semi-rigorous after all so we have deleted this text. Ideally we would be able to compare overlapping techniques at least semi-rigorously and often we have in previous studies. For instance, most recently in Hatch et al. (2016), the WAS vs FTIR slope for overlap species that were nearly the same as in this study was $1.01 \pm 0.001$, $r^2 = 1.0$. Unfortunately, the resources we could import to the field were limited and we did not perform conclusive tests. We have made the following changes to the text.

Old text: "There were nine instances when the same gas was measured by both WAS

and FTIR in nearly the same place and seven of these nine cases agree within the combined uncertainty. The other two cases are less close, but this experiment was not well-designed for comparison. We have noted excellent WAS/FTIR agreement previously under more rigorous, but drier conditions (Christian et al., 2003) and we found that these 2015 field WAS results compared well with on-line measurements during FLAME-4 peat fire sampling for many major species as discussed later in the paper."

New text: "This experiment was not well-designed for comparison, but we have noted excellent WAS/FTIR agreement previously under more rigorous, but drier conditions (e.g. Christian et al., 2003; Hatch et al., 2016) and we found that these 2015 field WAS results compared well with on-line measurements during FLAME-4 peat fire sampling for many major species as discussed later in the paper."

Hatch, L. E., Yokelson, R. J., Stockwell, C. E., Veres, P. R., Simpson, I. J., Blake, D. R., Orlando, J. J., and Barsanti, K. C.: Multi-instrument comparison and compilation of non-methane organic gas emissions from biomass burning and implications for smoke-derived secondary organic aerosol precursors, Atmos. Chem. Phys. Discuss., doi:10.5194/acp-2016-598, in review, 2016.

**R2.7a:** Page 12, lines 33-35 – many of the uncertainties are unreasonably precise – i.e., 0.867 ± 0.479 and 0.860 ± 0.433. Please round these to make them more reasonable for reporting.

**Authors:** We have rounded uncertainties in cases where the level of significance did not match (e.g. propylene 1.07 ± 0.531 now changed to 1.07 ± 0.53). However we like to report our results with any potentially useful amount of digits for several reasons that include minimizing round-off error if the number is used by others. The variability should be propagated for all terms used in any calculations done with these numbers and reported along with the result to avoid misleading uncertainties.

**R2.7b:** Page 13, lines 2-3 the "overlap" isn't very surprising, considering that the range

in your work is from 0.3 to 1.44. Overlap isn't hard, and likely shouldn't be emphasized like this. "Are consistent", perhaps.

**Authors:** We did not mean to imply overlap between a variable data set and a single point is difficult, but it is a typical reality check in comparing sets of data. A lack of overlap would imply serious questions about the relevance. We changed: "which overlaps with" to "roughly consistent with" – hopefully the point that significant emissions of large alkanes likely occur from real peat fires is made.

**R2.8:** Page 13, lines 10-15 – I'm not fond of the idea of alluding to something that should be done, and then just saying "we haven't attempted this yet." Why bring it up? Or why not attempt to include the analysis here?

**Authors:** Referee #1 also commented on this. Our thought in including this was that the reader would wonder if peat characteristics correlated with emissions and that therefore mapping peat characteristics would improve estimates. We wanted to point out that we don't see a way forward. However, as the Referee points out we can just delete it this text, which is what we have done.

**R2.9:** Page 14, lines 5-16 – I have issues with this plot, and with the implication that the overlap in time is so fortuitously going to take something with 7 points, eliminate 3, and leave you with a four point plot that has an r^2 of 0.674, and that you're going to give it any actual credence. I don't think you "confirmed" the MAC near 0.1 at all. You just eliminated points until the remainder of your points came slightly close to giving you a line. This either needs far more justification, or it shouldn't be included.

**Authors:** Figure 3 shows that the emissions may change with time so looking at the ratio of absorption to mass from data collected at the exact same time is important to consider and gives the value from the plot of $0.071 \pm 0.03$. If we relax our time-overlap restriction to tap into more data and compare all data as a straight average (a plot would have low $r^2$) we get nearly the same number $0.09 \pm 0.08$. These values overlap within uncertainty and because the mass measurement has a larger size cut-off, both

approaches imply a MAC value "near 0.1" – note that in this case we restrict ourselves to one significant figure as we are not quoting a measured uncertainty. As with the MSE estimate, we get similar values using just the perfectly overlapped data as we do with using all the data. To remind the reader why we considered temporal overlap in a dynamic environment we made the following change:

Old text: "If instead we plot EF $B_{abs-405}$ versus EF OC just for the four plumes sampled over the exact same time period (but different size cutoffs; blue points in Fig. 6) we get a slope of 0.071 $\pm$ 0.03 m$^2$/g."

New text: "Keeping the dynamic nature of the emissions chemistry shown in Fig. 3 in mind, if we restrict our analysis to the same four plumes where sample timing was identical (but different size cutoffs; blue points in Fig. 6) and plot EF $B_{abs-405}$ versus EF OC we get a slope of 0.071 $\pm$ 0.03 m$^2$/g."

**R2.10:** Page 14, line 36 – please comment on the differences between the Liu et al. paper using SSA 781 and you using SSA 870 nm, and what kind of linearity you expect for the two different locations, and how that affects your comparison. Also, this comparison of the observed aerosol parameters with literature values would benefit from having a table like S3 included.

**Authors:** At both 781 and 870, the absorption by BrC should be very small so the high SSA is consistent with minimal BC emissions. The amount of data available to compare here is limited and so we have elected to present this in the text rather than adding a very small table in the supplement. We did reformat the text to make it easier to read.

Old text: "Turning to optical properties, Liu et al. (2014) reported some SSA values and the AAE for smoldering Kalimantan peat (Fire 114) from FLAME-4: MCE (0.74), SSA 405 (0.94), SSA 781 (1.00), and AAE (6.06). These are very consistent with our data (Table 2) and especially with our lowest MCE field sample: MCE (0.726), SSA 405 (0.941), SSA 870 (0.997), and AAE (6.23)."

New text: "Turning to optical properties, Liu et al. (2014) reported some SSA values and the AAE for smoldering Kalimantan peat (Fire 114) from FLAME-4: MCE (0.74), AAE (6.06), SSA-405 (0.94), and SSA-781 (1.00). These are very consistent with our data (Table 2) and especially with our lowest MCE field sample: MCE (0.726), AAE (6.23), SSA-405 (0.941), and SSA-870 (0.997)."

The requested comment on the comparison was added a few lines further down: "BrC absorption is very small at both 781 and 870 nm so the high SSA at the long wavelengths in both studies and similar AAEs are consistent with minimal BC absorption and dominant absorption by BrC."

**R2.11:** Page 15, line 36 – "and other factors" is very vague. Please expound.

Authors: Quantifying the amount of fire emissions is extremely challenging, especially for peat fires and in SE Asia, as discussed at length in the cited reference by Reid et al. In bottom-up approaches, fires and burned area are missed due to cloud cover, which approaches 90% on average in SE Asia and also due to an aggressive "cloud mask" that rejects smoky pixels. MODIS only scans areas near the equator 2 out of every 3 days and at the edges of the scans the resolution is degraded to about 6 km, which misses numerous small, smoldering peat fires even if clouds (and canopy) don't interfere. In top-down (inverse modeling) approaches, fires and burned area can be underestimated due to the same factors above and general uncertainty is added due to problems such as uncertainty in air mass factors, emission factors, smoke injection altitude, meteorology, and the evolution of species used as constraints. Thus, most often the initial amount of fire emissions in inventories needs to be increased in models by factors of ∼1.5-10 to match observations. Some of many examples of this adjustment procedure include Lu and Sokolik (2013), Reddington et al. (2016), and extensive references cited within these papers. Rather than a long digression in the paper, we have edited this sentence and added more references showing that a-priori fire emissions are usually too low.

Old text: On the other hand, burned area is likely underestimated in inventories since they rely on remote sensing data that misses hotspots, burned area, and the fire products used in top-down approaches. This is due to high regional cloud cover; orbital gaps; rapid green-up, which is strongly associated with shallow burn depth (Cypert, 1961; Kotze, 2013); and other factors (Reid et al., 2013). Thus, overestimating burn depth and underestimating burned area tend to cancel.

New text: On the other hand, burned area is likely underestimated in inventories since they rely on remote sensing data that misses some of the hotspots and burned area used in bottom-up estimates, as well as some of the fire products (e.g. CO, aerosol) used in top-down approaches. The information gap is caused by high regional cloud cover; orbital gaps; rapid growth of new vegetation, which is strongly associated with shallow burn depth (Cypert, 1961; Kotze, 2013); and other factors (Lu and Sokolik, 2013; Reddington et al., 2016; Reid et al., 2013). Thus, overestimating burn depth and underestimating burned area tend to cancel when coupling these terms to estimate fuel consumption.

Lu, Z., and Sokolik, I. N.: The effect of smoke emission amount on changes in cloud properties and precipitation: A case study of Canadian boreal wildfires of 2007, J. Geophys. Res., 118, 11777–11793, doi:10.1002/2013JD019860, 2013.

Reddington, C. L., Spracklen, D. V., Artaxo, P., Ridley, D., Rizzo, L. V., and Arana, A.: Analysis of particulate emissions from tropical biomass burning using a global aerosol model and long-term surface observations, Atmos. Chem. Phys. Discuss., doi:10.5194/acp-2015-967, in review, 2016.

**R2.12:** Page 16, line 13 – you "interpolated" between two points to find something. How did you do this? Was it linear? Why? How do you know?

**Authors:** We don't have any data or theory to support anything other than a simple linear interpolation and we now specify that this was what we did.

Old text: This is close to our 530 nm value if we interpolate between 870 and 405 nm (0.981).

New text: We can estimate an SSA at 530 nm by linear interpolation between 870 and 405 nm and obtain a similar value (0.981).

**R2.13:** Page 15-16, Section 3.3. This section feels very hand-wavy. I would like to see a more quantitative and step-wise analysis presented for the comparisons mentioned to previous studies and other kinds of peat BB observations in this paragraph. Some of the comparisons mentioned are presented with little defense as to their relevance and/or the validity of the comparison (i.e., the interpolation mentioned above.)

**Authors:** There are very little data to compare to so we compared to everything with any relevance. In addition, we have now added comparisons to two more papers that came to our attention after we submitted our paper (and we updated the text and Table S3 to reflect this). Hopefully by addressing the specific comments and adding these new comparisons this section will be more useful. Even if we cannot address the comparisons in this section as conclusively as might be liked, representativeness is an important issue and we have attempted to raise and discuss it to the extent possible.

P16, L30 (end of section 3.3) added text: "Two very recent studies probed peat fire emissions during the 2015 El-Niño. Huijnen et al. (2016) measured three EFs for peat fires also near Palangkaraya. Their "peat-only" EFs are 255 $\pm$ 39, 1594 $\pm$ 61, and 7.4 $\pm$ 2.3 g/kg for CO, $CO_2$ and $CH_4$, respectively. Their means are all within one standard deviation of our means and their EFs are within +1.9, -13, and -22% percent of ours, respectively. Not many details of the measurements are given, but the agreement is good. Parker et al. (2016) report three space-based measurements of the ER for Kalimantan fires in Sept-Oct 2015 for $CH_4/CO_2$ ranging from 0.0062 to 0.0136. This is lower on average than the $CH_4/CO_2$ ERs reported for peat combustion in the in-situ studies cited above (range $\sim$0.011 – 0.035). The difference is consistent with our expectation noted above that some flaming-dominated consumption of surface

fuels likely contributed to regional emissions in 2015. However, a glance at Figure 6 in Parker et al. (2016) shows that some of highest retrieved levels of these gases, which they attribute to fires, are far off-shore and/or upwind of the fires. Thus, more evaluation is clearly needed to determine if space-based approaches can accurately measure $CH_4/CO_2$ ERs (e.g. Agustí-Panareda et al., 2016)."

Agustí-Panareda, A., Massart, S., Chevallier, F., Balsamo, G., Boussetta, S., Dutra, E., and Beljaars, A.: A biogenic $CO_2$ flux adjustment scheme for the mitigation of large-scale biases in global atmospheric $CO_2$ analyses and forecasts, Atmos. Chem. Phys., 16, 10399-10418, doi:10.5194/acp-16-10399-2016, 2016.

Huijnen, V., Wooster, M. J., Kaiser, J. W., Gaveau, D. L. A., Flemming, J., Parrington, M., Inness, A., Murdiyarso, D., Main, B., and van Weele, M.: Fire carbon emissions over maritime southeast Asia in 2015 largest since 1997, Scientific Reports, 6, 26886, doi:10.1038/srep26886, 2016.

Parker, R. J., Boesch, H., Wooster, M. J., Moore, D. P., Webb, A. J., Gaveau, D., and Murdiyarso, D.: Atmospheric $CH_4$ and $CO_2$ enhancements and biomass burning emission ratios derived from satellite observations of the 2015 Indonesian fire plumes, Atmos. Chem. Phys., 16, 10111-10131, doi:10.5194/acp-16-10111-2016, 2016.

**R2.14:** Page 17, line 33 – "the lab value is actually the sum of isomers compared to a single isomer from the GC analysis: : :" please explain this more, including references to the table, in which I see no evidence of a difference between a sum of isomers and a single isomer. Is this for a particular compound or set of compounds? Be specific.

**Authors:** This is a very helpful and important comment. One thing that was clear in the FLAME-4 papers, but we failed to specify in this paper was that the mass spec assignments in the lab study were "nominal" in some cases: or in other words a best guess at the most abundant species when many isomers could contribute. For example, mass 137.132 in the lab studies was calibrated with alpha-pinene and shown as that compound in Table S3, but there are numerous isomers that have the same exact

mass (Hatch et al., 2015; 2016). In addition, fragments of higher masses can contribute to the mass spec signals. Thus it is not surprising that the lab MS value for all of mass 137.132 is much larger than the field WAS value more specific to alpha-pinene. In addition, one of the FLAME-4 PTR-TOF-MS calibrations was just revised based on the work described in Hatch et al., (2016). Thus, we have revisited this in detail. We have revised the text and the species impacted by isomers are now clearly flagged in Table S3.

P17, L30: Old text: "Table S3 compares all 31 gases measured for Kalimantan samples in both the lab (FLAME-4) and the field. The average of the two lab EFs is within a factor of two of the field mean for 20 of 31 species, which is adequate given that a factor of two is essentially also the field coefficient of variation ($n = 35$). In 7 of the 11 cases with more than a factor of two difference, the lab value is actually the sum of isomers compared to a single isomer from the GC analysis of the field WAS samples. For the remaining 4 species the lab values tend to be higher for unclear reasons. For instance formic acid is higher in the lab where an open-path system was used instead of the closed cell system in the field, which could be subject to sample losses. However, HCl and $NH_3$ are likely more prone to adsorption than formic acid (Yokelson et al., 2003) and they were higher as measured with the field system suggesting the Teflon sample line and coating on the closed cell were effective in limiting line losses. The lab average for NO is higher, but NO was below detection in one lab fire and high in the other where flaming briefly occurred. The one field fire where flaming was briefly observed (Plume C, Table S2) had a higher EF for NO than the lab fire where it was detected. Thus, further comparisons with more lab fires will clearly be useful, but it appears the trace gas EFs from the lab are reasonable proxies for the EFs for species that have not been measured in the field"

New text: "Table S3 compares all 31 gases nominally measured for Kalimantan samples in both the lab (FLAME-4) and the field. (We clarify the need for the term "nominal" below.) Due to the natural high variability in the field data, the low number of lab measurements (two), the use of different peat samples, etc., we start by proposing that the lab measurements provide useful EFs for species not measured in the field if the average of the two lab EFs is within a factor of two of the field mean for species measured in both locations. Next, we find in the right-hand column of Table S3 that 15 of 31 species fail this initial factor-of-two test (ratios shown in red). However, this result is somewhat misleading since the lab data for 8 of these species (shown in blue) is actually comparing a best guess at the identity of the most abundant isomer for an exact mass measured in the lab to a WAS-based analysis for a specific isomer. Thus, these ratios could be larger than two because of contributions from other isomers (or fragments) to the mass spectrometer signal, higher than normal sensitivity in the mass spectrometer, WAS error, or unusually high variability for some species; with no way of knowing the individual contribution of these factors. We do note that a generally good comparison of the WAS and mass spectrometer was obtained when they were compared more directly in peat smoke in the lab (Hatch et al., 2016). Thus, only 7 out of 23 compounds fail the factor-of-two test, if we eliminate species that are ambiguous due to isomers. Of these 7 species, three are very close to the factor-of-two cutoff and are of less concern (ammonia, acetaldehyde, and hydroxyacetone). For the remaining 4 species (formic acid, NO, 1,3-butadiyne, styrene) the lab values tend to be higher for unclear reasons. For instance formic acid was higher in the lab where an open-path FTIR system was used instead of the closed cell FTIR system in the field, which could be subject to sample losses. However, HCl (below detection in lab) and $NH_3$ are likely more prone to adsorption than formic acid (Yokelson et al., 2003) and they were higher as measured with the field system suggesting the Teflon sample line and coating on the closed cell were effective in minimizing losses and sampling losses were not the source of the discrepancy. The lab "average" for NO was more than four times higher than the field value implicating high variability. NO was below detection in one lab fire and "high" in the other lab fire where flaming briefly occurred. The one field fire where flaming was briefly observed (Plume C, Table S2) had an even higher EF for NO than in the lab fire where it was detected. The other two species of concern are styrene and

1,3-butadiyne. These two ratios could be high due to decay in the canisters, fragments in the mass spectrometer, or perhaps other less likely reasons. In summary, more lab/field comparisons should be carried out, but our rough analysis suggests that trace gas EFs measured in the lab are useful estimates (i.e. within a factor of two) for the emissions of most gases not yet measured in the field."

**R2.15:** Page 18, line 14 – you should reassure the reader that the 37% of unidentified or tentatively assigned mass peaks of the NMOG mass is not going to negatively affect your assumption that you are measuring all the carbon to be factored into your EF calculations. I'm sure it's not significant, considering the major non-NMOG carbon species, but this should be recognized.

**Authors:** P19 L 33-35 we added: "The missing NMOG mass in the field measurements is not large enough to cause significant error in our field carbon mass balance, but would impact estimates of secondary formation of aerosol and $O_3$ (Yokelson et al., 2013; Hatch et al., 2015)."

**R2.16:** In Table 1, Table 2, Figure 1, etc., there are numbers that are both too precise considering the standard deviations reported, and I dislike the excel-style presentation of numbers with exponents written as (e.g.) 1.67E-3.

**Authors:** We have changed the tables so they are no longer in the excel-type format. . The number of figures in the table values is to avoid round-off error, though we have changed values where the significant figures did not match: e.g. CH3I, 0.0125(0.00448) corrected to 0.0125(0.0045).

**R2.17:** Re: Table S1 – there are a handful of things that would make this table easier to digest, without having to search out other information.

**Authors:** We have italicized the Referees suggestions in the rest of this comment and reply point by point in un-italicized text.

*Instead of "Y/N type" or "Y/N what" as a header, eliminate Y/N and just include the*

*type/what or put "none" or "unknown" where applicable.*

Au: Done.

*Also, please spell out here what the peat fuel types are, so that I don't have to go back and find that in the paper (in the footnote would be fine.)*

Au: They are now spelled out in each cell.

*"day-mon" should be "DD-Mon".*

Au: Done.

*Why are there some plumes included that aren't lettered? These don't seem to add anything to the paper.*

Au: The unlettered plumes represent occasions when it was possible to quickly collect a WAS canister, but not deploy all the instruments. These samples add important emissions information.

*"seec" is not a word or a shortform (as a direction).*

Au: "seec" was changed to "sec"

*For Depth of Burn, " site avg" is redundant. "site avg" is fine.*

Au: We changed to "approx. site avg" and deleted the "$\sim$" where they appeared before entries.

*Why is so little known about site 6?*

Au: The team member collecting site data was unable to participate on the last day of the field campaign. The emissions team was able to collect some data.

*Re: Winds – "av, max, dir" implies you'll have numeric values below. Maybe leave the "av, max, dir" part out, and just consider it a verbal description of the winds.*

Au: Done.

*Be consistent with spacing and vertical cell centering.*

Au: This was checked.

*Also, for all three supporting information tables (S1-S3), please be consistent about font size and styles and remove bold settings.*

Au: The journal has requested that we use 9 pt Times New Roman with bold headings for our print tables and we assumed they want that for the supplement as well. However, Table S1 was clearly outside the normal journal format so we changed the font and font size to optimize readability.

*Table titles should all be uniformly sized.*

Au: Please see above.

*For all supplement tables, if these are being submitted as they are now in an excel file, a san serif font is likely best for readability. If you're preparing a printed document, a simple serif font is also acceptable (i.e., Times New Roman.)*

Au: We have been asked to use Times New Roman 9 pt for tables by the journal so we did that. On Table S1 we used a more readable font since it is clearly a non-standard table.

**Referee #2 (Continued) Technical Comments**

**R2.18:** Page 5, lines 4-5 – don't us semicolons in place of commas.

**Authors:** We separated the items in the list with semi colons because one item has multiple terms separated by commas: ". . . ; charred logs, char, and ash from previous burns;"

**R2.19:** Page 5, line 26 – remove the hyphen from "at six-different peatland: : :"

**Authors:** removed

**R2.20:** Page 6, line 10 – no need for "(#2)" after "This site: : :"

**Authors:** deleted

**R2:21:** Page 11, line 12 – there is a missing or extra parenthesis here.

**Authors:** The first "(" was deleted and now reads "$1/(1+\Delta CO/\Delta CO_2)$"

**R2.22:** Page 18, line 28 – "$0.35 \pm .1$ x 10: : :" – the .1 should be 0.10.

**Authors:** fixed

**R2.23:** Page 19, line 1 – "Six of the nine: : :"

**Authors:** changed to "six of the eleven"

**R2.24:** Page 19, line 11 – the Putra et al. paper in preparation needs to be included in the reference list.

**Authors:** added

**R2.25:** Tables 1, S2 and S3: "ethyne", "ethene", "propene" (and in the text and Figure 1, where applicable).

**Authors:** We think the common name for these species and many others (e.g. formaldehyde, formic acid, methanol and many more) is more or reasonably common, but added the requested formal names in the parentheses with the formula at the first appearance of these species in the text (P7).

---

## Author Response (AR1)

Dear Delphine: Thank you for your hard work. Our posted responses to all comments are followed by the complete manuscript with all revisions shown in track changes. Thanks again! Bob

**Response to Referee #1**

5 We thank the Referee for their encouraging assessment and constructive suggestions, which will improve the paper. The Referee comments are reproduced below followed by our detailed response.

**Anonymous Referee #1**

30

The manuscript "Field measurements of trace gases and aerosols emitted by peat fires in Central Kalimantan, Indonesia during the 2015 El Niño" presents the first field emission measurements of comprehensive atmospheric

- 10 compositions from peat fires burning in Southeast Asia. This kind of field measurements is extremely rare and thus very valuable to the scientific community of Atmospheric Chemistry and Physics. The measurement methods used in the study are well established and the field experiment design is reasonable and justified. I expected this manuscript would only reported emissions from a rarely studied environment (which itself would add values to literature), but the discussion on the representativeness of the field measurements, comparison to previously
- 15 available emission factors for the same type of emissions is very useful too. The authors also compare the field measured emission factors to those obtained from lab experiments, and discuss the value and importance of lab data. Peat fire burning in Southeast Asia is such an interesting and important topic from atmospheric chemistry and climate perspectives but many questions still remain as first order research problems due to the limited field data. I believe the manuscript could be much improved in terms of how to scale the field data to a large spatial area in this
- 20 region, but I understand that the study is also limited by prior data and resource that could be deployed. The manuscript is well written in general, while the readability could be improved by properly introducing acronyms. In summary, I think this manuscript could be published and I list a few minor suggestions as below:

R1.1: The manuscript points the importance and uniqueness of 2015 El Niño event. The authors need to comment on how this field measurements during a El Niño event apply to other 'normal' years, or do the authors suggest that these field measured emission factors can only apply to El Niño events? Can the difference between lab and field comparison be partly explained by the special El Niño event?

**Authors:** This is an excellent question. The emissions are "possibly" of greatest interest during El Niño years when the "acute" impacts are greatest. For now we can only assume that the emission factors for burning peat (g/kg peat burned) are probably similar in all years, but the total emissions (Tg/yr) are smaller in non-El Niño years when the fire season is not as severe and the amount of peat burned is reduced. However, the summed emissions from several

- non-El Niño years will surely rival the emissions from an El Niño year. We don't know enough about what drives the variability in peat-fire emission factors at this point to speculate whether El Niño will drive interannual variability in peat-fire emission factors or lab/field differences, but we do plan future measurements designed in part to probe geographic and inter-annual differences. We're not sure if we should address these questions in this paper,
- 35 but we now clarify that despite the recent increase in non-El-Niño year emissions, the El Niño year emissions are still anomalously large at P3, L19:

Old text: "With accelerated deforestation and building of drainage canals (e.g. 4000 km of canals as part of the Ex Mega Rice Project (EMRP) started in 1996 (Putra et al., 2008; Hamada et al., 2013)), peat fires and their impacts are now extensive on an annual basis (van der Werf et al., 2010; Wiedinmyer et al., 2011; Gaveau et al., 2014)."

40 New text: "With accelerated deforestation and building of drainage canals (e.g. 4000 km of canals as part of the Ex Mega Rice Project (EMRP) started in 1996 (Putra et al., 2008; Hamada et al., 2013)), peat fires and their impacts are now extensive on an annual basis (van der Werf et al., 2010; Wiedinmyer et al., 2011; Gaveau et al., 2014) and even more pronounced in El Niño years (Huijnen et al., 2016)."

Regarding the difference between lab and field results, while the 2015 El Niño explains larger overall emissions, the lab/field comparison is based on emission factors (g/kg) relative to units of peat burned, and as a result we believe the comparison should be useful.

**R1.2:** Related to point 1: This manuscript finds that many significant revisions of emission factors compared previously widely used EFs, mostly reductions (CO2, CH4, NH3). But as the authors point in the introduction, previous studies suggest "in Southeast Asia, in the 1980s-1990s, peatland fires were a major source of carbon to the atmosphere mainly during El Niño induced droughts : : .: " How can the authors reconcile this? The manuscript uses

10 "the 2015 El Niño" in the title, and the authors would be expected to comment more on this event. However, such comments are very rare in this version of the manuscript.

Authors: We think the Referee is wondering if we mentioned El Niño in the title because we think El-Niño changes the chemical nature of emissions. Our inclusion of El Nino in the title was just to point out the data was collected in a year responsible for a large fraction of the total emissions from the region on a multi-year time-scale. I.e. we went

in a year when the conditions were associated with more emissions than most other years. This doesn't prove the data is more relevant, but it is a good choice in case the peat-fire emission factors do change inter-annually. Hopefully this is clear now that we clarify (as described above) that the largest "acute" impacts occur in El Niño years.

R1.3: P5 L14: the definition of fuel moisture is not clear. What is 'wet', what is 'dry'? Here and many places in the
manuscript, the authors assume all readers know most of acronyms related to fire studies. Properly introducing them
could help the manuscript reach a broad audience of atmospheric scientists.

Authors: Thank you for this comment. We have changed text to read: "Peat deposits can burn at > 100% fuel moisture (defined as  $100 \times (wet-dry)/dry$ )), where "wet" refers to the weight of a fresh fuel sample and "dry" refers to the fuel weight after oven drying until mass loss ceases. This is because the glowing front pre-dries the fuel as it advances."

25

5

**R1.4:** P7 L29: 'cyclones' should be 'cyclone samplers'? The authors need to avoid using 'field language' as much as possible and try to use its formal name.

Authors: Changed (P7, L31 in the revised text).

**R1.5:** P8 L15: poorly written.

30 **Authors:** We made this more definitive by changing to: "Styrene is known to decay in canisters and the styrene data should be taken as lower limits."

**R1.6:** P8 L21-22: here and other places, the instrument modes and manufactures should be listed as full names with company names and locations.

Authors: We added this information here and in many other places. In just a few cases we skipped to avoid overly long sentences since the supplied info is enough to locate further info on the web.

**R1.7:** P 9, session 2.2.5: it is unclear if any control (or blank) samples were deployed for these offline measurements? For example, pre-cleaned filters shipped with other filters but without any sampling.

Authors: We did collect field blanks, but since we elected to subtract background filters from source sample filters the field blank correction cancels mathematically. I.e. S-B = (S-c) - (B-c) = S-c-B+c. At P10, L10 we added:

"While field blanks were collected, subtracting the background from smoke samples made the field blank correction unnecessary."

**R1.8:** P13 L13-15: it would be very valuable if those peat characteristics were mapped and it could help to scale these point measurements to large areas and perhaps devise a parameterization to study other peat fire emission. It is very unfortunate that this study did not attempt such an analysis. What would be the authors' recommendations to future field studies? It would be useful for other researchers who are interested in this area.

Authors: We agree this is a logical, important thing to wonder about and that is why we mentioned it. However, we are not sure if there are peat characteristics that correlate with emission differences. We also don't know how well characteristics such as peat type, moisture, etc could be mapped in 3-D along with burn depth. If these connections

10 exist and could be mapped/scaled it would require a substantial additional study. Another Referee questioned the relevance of this inconclusive text so we ultimately deleted it.

**R1.9:** P15 L 36: what is 'rapid green-up'. Again, the manuscript could be improved and reach a broader audience if these words were properly defined.

Authors: "Green-up" is a forestry term to describe the emergence of fresh new vegetative growth, which is "green"
rather than the "black" as might be expected for a burn scar, or "brown" due to a drought, etc. We changed "rapid green-up," to "rapid growth of new vegetation,"

**Response to Referee #2**

We thank the Referee for their encouraging assessment and constructive suggestions, which will improve the paper. The Referee comments are reproduced below followed by our detailed response.

20 Anonymous Referee #2

5

Review of "Field measurements of trace gases and aerosols emitted by peat fires in Central Kalimantan, Indonesia during the 2015 El Niño", by C. E. Stockwell et al., 2016. The manuscript by Stockwell et al. presents measurements from 2015 peatland fires in the Indonesian province Kalimantan. The findings presented in this manuscript both add to and modify previous lab-based measurements of peat combustion, amending a handful of

25 key EFs that were previously only available from laboratory studies, while confirming the validity of laboratory studies for estimating EFs for species that are not easily measured in the field. The paper is well-written, cohesive and thorough, and my assessment is that it merits publication in ACP after the following issues are addressed.

Specific Comments

**R2.1:** Page 2, line 19 – "(2012)" - is that a reference? It should be included properly here.

30 **Authors:** We found "(2012)" on P2, L8. The year of the recent measurements was included because our field data did not agree well with the lab measurements made in 2001 that IPCC uses, but did agree well with lab measurements made in 2012. It's probably safe to delete the year and we have done so.

R2.2: Page 4, line 32 and Page 19, line 10 – I don't think it's necessary to put "in preparation" here – it is included
in the reference itself. However, please include a full reference for this work if possible, including a full author list and title.

Authors: We removed "in preparation" throughout and updated the reference as suggested.

**R2.3:** Page 5, lines 11-12 – please explain what n=1 is in reference to, or simply state, if this is the case, that there was only a single sample of each type analyzed in the lab study.

Authors: Within both parentheses we changed "n=1" to "one sample"

**R2.4:** Page 5, line 33 – it would be helpful to the reader to direct them to Table S1 at this point, rather than making them wait to find out about the table until the next page. **Authors:** Good suggestion and we now call out Table S1 earlier on P5, L27.

5 **R2.5:** Page 6, lines 4-25 – a map/diagram of the sites would help put this entire sampling description into context. **Authors:** A site map has now been provided as Supplementary Figure 2 (now referenced on P6 L4).

**R2.6:** Page 12, lines 20-25 – where are the ERs if I want to look at them? A lot of this discussion is very qualitative and vague (i.e. "seven of these nine cases agree: ::" – what about the other two? And what were they? It would

- 10 seem reasonable to spell that out or offer something specific about the differences between the FTIR and the WAS sample that would alleviate the reader's concern that there is something we should know about the alleged differences. I understand that the analysis were not set up to evaluate differences, and yet to just allude to it but not give us anything further is more suspicious.
- Authors: This is an excellent comment and we considered adding this comparison to the supplement in response.
   For background, we had two WAS cans that the field notes indicated were filled from the FTIR cell. While changes could theoretically occur to some species during storage in the FTIR cell, this seemed like a valid opportunity to compare the data for the overlap species fairly directly. Unfortunately, in attempting to further clarify this comparison, we have now realized that the FTIR cell was actually re-filled after the FTIR measurements and before
- the cans were filled. Thus, the comparison included an unknown contribution from natural plume variability and was not semi-rigorous after all so we have deleted this text. Ideally we would be able to compare overlapping techniques at least semi-rigorously and often we have in previous studies. For instance, most recently in Hatch et al. (2016), the WAS vs FTIR slope for overlap species that were nearly the same as in this study was  $1.01 \pm 0.001$ ,  $r^2 = 1.0$ . Unfortunately, the resources we could import to the field were limited and we did not perform conclusive tests. We have made the following changes to the text.
- 25

Old text: "There were nine instances when the same gas was measured by both WAS and FTIR in nearly the same place and seven of these nine cases agree within the combined uncertainty. The other two cases are less close, but this experiment was not well-designed for comparison. We have noted excellent WAS/FTIR agreement previously under more rigorous, but drier conditions (Christian et al., 2003) and we found that these 2015 field WAS results compared well with on line measurements during FLAME 4 neat fire sampling for many major species as discussed

30 compared well with on-line measurements during FLAME-4 peat fire sampling for many major species as discussed later in the paper."

New text: "This experiment was not well-designed for comparison, but we have noted excellent WAS/FTIR agreement previously under more rigorous, but drier conditions (e.g. Christian et al., 2003; Hatch et al., 2016) and we found that these 2015 field WAS results compared well with on-line measurements during FLAME-4 peat fire sampling for many major species as discussed later in the paper."

Hatch, L. E., Yokelson, R. J., Stockwell, C. E., Veres, P. R., Simpson, I. J., Blake, D. R., Orlando, J. J., and Barsanti, K. C.: Multi-instrument comparison and compilation of non-methane organic gas emissions from biomass burning and implications for smoke-derived secondary organic aerosol precursors, Atmos. Chem. Phys. Discuss., doi:10.5194/acp-2016-598, in review, 2016.

**R2.7a:** Page 12, lines 33-35 - many of the uncertainties are unreasonably precise -i.e.,  $0.867 \pm 0.479$  and  $0.860 \pm 0.433$ . Please round these to make them more reasonable for reporting.

- 45 Authors: We have rounded uncertainties in cases where the level of significance did not match (e.g. propylene 1.07  $\pm$  0.531 now changed to 1.07  $\pm$  0.53). However we like to report our results with any potentially useful amount of digits for several reasons that include minimizing round-off error if the number is used by others. The variability should be propagated for all terms used in any calculations done with these numbers and reported along with the result to avoid misleading uncertainties.
- 50

40

**R2.7b:** Page 13, lines 2-3 the "overlap" isn't very surprising, considering that the range in your work is from 0.3 to 1.44. Overlap isn't hard, and likely shouldn't be emphasized like this. "Are consistent", perhaps. **Authors:** We did not mean to imply overlap between a variable data set and a single point is difficult, but it is a typical reality check in comparing sets of data. A lack of overlap would imply serious questions about the relevance.

55 We changed: "which overlaps with" to "roughly consistent with" – hopefully the point that significant emissions of large alkanes likely occur from real peat fires is made.

**R2.8:** Page 13, lines 10-15 – I'm not fond of the idea of alluding to something that should be done, and then just saying "we haven't attempted this yet." Why bring it up? Or why not attempt to include the analysis here? **Authors:** Referee #1 also commented on this. Our thought in including this was that the reader would wonder if

- 5 peat characteristics correlated with emissions and that therefore mapping peat characteristics would improve estimates. We wanted to point out that we don't see a way forward. However, as the Referee points out we can just delete it this text, which is what we have done.
- R2.9: Page 14, lines 5-16 I have issues with this plot, and with the implication that the overlap in time is so
  fortuitously going to take something with 7 points, eliminate 3, and leave you with a four point plot that has an r2 of 0.674, and that you're going to give it any actual credence. I don't think you "confirmed" the MAC near 0.1 at all. You just eliminated points until the remainder of your points came slightly close to giving you a line. This either needs far more justification, or it shouldn't be included.
- Authors: Figure 3 shows that the emissions may change with time so looking at the ratio of absorption to mass from data collected at the exact same time is important to consider and gives the value from the plot of  $0.071 \pm 0.03$ . If we relax our time-overlap restriction to tap into more data and compare all data as a straight average (a plot would have low r2) we get nearly the same number  $0.09 \pm 0.08$ . These values overlap within uncertainty and because the mass measurement has a larger size cut-off, both approaches imply a MAC value "near 0.1" – note that in this case we restrict ourselves to one significant figure as we are not quoting a measured uncertainty. As with the MSE estimate,
- 20 we get similar values using just the perfectly overlapped data as we do with using all the data. To remind the reader why we considered temporal overlap in a dynamic environment we made the following change: Old text: "If instead we plot EF  $B_{abs-405}$  versus EF OC just for the four plumes sampled over the exact same time period (but different size cutoffs; blue points in Fig. 6) we get a slope of  $0.071 \pm 0.03 \text{ m}^2/\text{g}$ ." New text: "Keeping the dynamic nature of the emissions chemistry shown in Fig. 3 in mind, if we restrict our
- 25 analysis to the same four plumes where sample timing was identical (but different size cutoffs; blue points in Fig. 6) and plot EF  $B_{abs-405}$  versus EF OC we get a slope of  $0.071 \pm 0.03 \text{ m}^2/\text{g.}$ "

**R2.10:** Page 14, line 36 – please comment on the differences between the Liu et al. paper using SSA 781 and you using SSA 870 nm, and what kind of linearity you expect for the two different locations, and how that affects your

30 comparison. Also, this comparison of the observed aerosol parameters with literature values would benefit from having a table like S3 included.
Authors: At both 781 and 870, the absorption by BrC should be very small so the high SSA is consistent with minimal BC emissions. The amount of data available to compare here is limited and so we have elected to present this in the text rather than adding a very small table in the supplement. We did reformat the text to make it easier to

35 read.

Old text: "Turning to optical properties, Liu et al. (2014) reported some SSA values and the AAE for smoldering Kalimantan peat (Fire 114) from FLAME-4: MCE (0.74), SSA 405 (0.94), SSA 781 (1.00), and AAE (6.06). These are very consistent with our data (Table 2) and especially with our lowest MCE field sample: MCE (0.726), SSA 405 (0.941), SSA 870 (0.997), and AAE (6.23)."

New text: "Turning to optical properties, Liu et al. (2014) reported some SSA values and the AAE for smoldering Kalimantan peat (Fire 114) from FLAME-4: MCE (0.74), AAE (6.06), SSA-405 (0.94), and SSA-781 (1.00). These are very consistent with our data (Table 2) and especially with our lowest MCE field sample: MCE (0.726), AAE (6.23), SSA-405 (0.941), and SSA-870 (0.997)."

The requested comment on the comparison was added a few lines further down: "BrC absorption is very small at both 781 and 870 nm so the high SSA at the long wavelengths in both studies and similar AAEs are consistent with minimal BC absorption and dominant absorption by BrC."

50

40

45

**R2.11:** Page 15, line 36 – "and other factors" is very vague. Please expound. Authors: Quantifying the amount of fire emissions is extremely challenging, especially for peat fires and in SE Asia, as discussed at length in the cited reference by Reid et al. In bottom-up approaches, fires and burned area are missed due to cloud cover, which approaches 90% on average in SE Asia and also due to an aggressive "cloud mask" that

rejects smoky pixels. MODIS only scans areas near the equator 2 out of every 3 days and at the edges of the scans the resolution is degraded to about 6 km, which misses numerous small, smoldering peat fires even if clouds (and

canopy) don't interfere. In top-down (inverse modeling) approaches, fires and burned area can be underestimated due to the same factors above and general uncertainty is added due to problems such as uncertainty in air mass factors, emission factors, smoke injection altitude, meteorology, and the evolution of species used as constraints. Thus, most often the initial amount of fire emissions in inventories needs to be increased in models by factors of

- 5 ~1.5-10 to match observations. Some of many examples of this adjustment procedure include Lu and Sokolik (2013), Reddington et al. (2016), and extensive references cited within these papers. Rather than a long digression in the paper, we have edited this sentence and added more references showing that a-priori fire emissions are usually too low.
- 10 Old text: On the other hand, burned area is likely underestimated in inventories since they rely on remote sensing data that misses hotspots, burned area, and the fire products used in top-down approaches. This is due to high regional cloud cover; orbital gaps; rapid green-up, which is strongly associated with shallow burn depth (Cypert, 1961; Kotze, 2013); and other factors (Reid et al., 2013). Thus, overestimating burn depth and underestimating burned area tend to cancel.

15

30

35

40

55

New text: On the other hand, burned area is likely underestimated in inventories since they rely on remote sensing data that misses some of the hotspots and burned area used in bottom-up estimates, as well as some of the fire products (e.g. CO, aerosol) used in top-down approaches. The information gap is caused by high regional cloud cover; orbital gaps; rapid growth of new vegetation, which is strongly associated with shallow burn depth (Cypert,

20 1961; Kotze, 2013); and other factors (Lu and Sokolik, 2013; Reddington et al., 2016; Reid et al., 2013). Thus, overestimating burn depth and underestimating burned area tend to cancel when coupling these terms to estimate fuel consumption.

Lu, Z., and Sokolik, I. N.: The effect of smoke emission amount on changes in cloud properties and precipitation: A
 case study of Canadian boreal wildfires of 2007, J. Geophys. Res., 118, 11777–11793, doi:10.1002/2013JD019860, 2013.

Reddington, C. L., Spracklen, D. V., Artaxo, P., Ridley, D., Rizzo, L. V., and Arana, A.: Analysis of particulate emissions from tropical biomass burning using a global aerosol model and long-term surface observations, Atmos. Chem. Phys. Discuss., doi:10.5194/acp-2015-967, in review, 2016.

**R2.12:** Page 16, line 13 – you "interpolated" between two points to find something. How did you do this? Was it linear? Why? How do you know?

Authors: We don't have any data or theory to support anything other than a simple linear interpolation and we now specify that this was what we did.

Old text: This is close to our 530 nm value if we interpolate between 870 and 405 nm (0.981).

New text: We can estimate an SSA at 530 nm by linear interpolation between 870 and 405 nm and obtain a similar value (0.981).

**R2.13:** Page 15-16, Section 3.3. This section feels very hand-wavy. I would like to see a more quantitative and stepwise analysis presented for the comparisons mentioned to previous studies and other kinds of peat BB observations in this paragraph. Some of the comparisons mentioned are presented with little defense as to their relevance and/or the validity of the comparison (i.e., the interpolation mentioned above.)

- 45 the validity of the comparison (i.e., the interpolation mentioned above.) Authors: There are very little data to compare to so we compared to everything with any relevance. In addition, we have now added comparisons to two more papers that came to our attention after we submitted our paper (and we updated the text and Table S3 to reflect this). Hopefully by addressing the specific comments and adding these new comparisons this section will be more useful. Even if we cannot address the comparisons in this section as
- 50 conclusively as might be liked, representativeness is an important issue and we have attempted to raise and discuss it to the extent possible.

P16, L30 (end of section 3.3) added text: "Two very recent studies probed peat fire emissions during the 2015 El-Niño. Huijnen et al. (2016) measured three EFs for peat fires also near Palangkaraya. Their "peat-only" EFs are 255  $\pm$  39, 1594  $\pm$  61, and 7.4  $\pm$  2.3 g/kg for CO, CO2 and CH4, respectively. Their means are all within one standard deviation of our means and their EFs are within +1.9, -13, and -22% percent of ours, respectively. Not many details of the measurements are given, but the agreement is good. Parker et al. (2016) report three space-based measurements of the ER for Kalimantan fires in Sept-Oct 2015 for  $CH_4/CO_2$  ranging from 0.0062 to 0.0136. This is lower on average than the  $CH_4/CO_2$  ERs reported for peat combustion in the in-situ studies cited above (range ~0.011 – 0.035). The difference is consistent with our expectation noted above that some flaming-dominated

- 5 consumption of surface fuels likely contributed to regional emissions in 2015. However, a glance at Figure 6 in Parker et al. (2016) shows that some of highest retrieved levels of these gases, which they attribute to fires, are far off-shore and/or upwind of the fires. Thus, more evaluation is clearly needed to determine if space-based approaches can accurately measure  $CH_4/CO_2$  ERs (e.g. Agustí-Panareda et al., 2016)."
- 10 Agustí-Panareda, A., Massart, S., Chevallier, F., Balsamo, G., Boussetta, S., Dutra, E., and Beljaars, A.: A biogenic CO2 flux adjustment scheme for the mitigation of large-scale biases in global atmospheric CO2 analyses and forecasts, Atmos. Chem. Phys., 16, 10399-10418, doi:10.5194/acp-16-10399-2016, 2016.

Huijnen, V., Wooster, M. J., Kaiser, J. W., Gaveau, D. L. A., Flemming, J., Parrington, M., Inness, A., Murdiyarso,
D., Main, B., and van Weele, M.: Fire carbon emissions over maritime southeast Asia in 2015 largest since 1997,
Scientific Reports, 6, 26886, doi:10.1038/srep26886, 2016.

Parker, R. J., Boesch, H., Wooster, M. J., Moore, D. P., Webb, A. J., Gaveau, D., and Murdiyarso, D.: Atmospheric  $CH_4$  and  $CO_2$  enhancements and biomass burning emission ratios derived from satellite observations of the 2015 Indonesian fire plumes, Atmos. Chem. Phys., 16, 10111-10131, doi:10.5194/acp-16-10111-2016, 2016.

**R2.14:** Page 17, line 33 - "the lab value is actually the sum of isomers compared to a single isomer from the GC analysis: : :" please explain this more, including references to the table, in which I see no evidence of a difference between a sum of isomers and a single isomer. Is this for a particular compound or set of compounds? Be specific.

25

20

**Authors:** This is a very helpful and important comment. One thing that was clear in the FLAME-4 papers, but we failed to specify in this paper was that the mass spec assignments in the lab study were "nominal" in some cases: or in other words a best guess at the most abundant species when many isomers could contribute. For example, mass 137.132 in the lab studies was calibrated with alpha-pinene and shown as that compound in Table S3, but there are

30 numerous isomers that have the same exact mass (Hatch et al., 2015; 2016). In addition, fragments of higher masses can contribute to the mass spec signals. Thus it is not surprising that the lab MS value for all of mass 137.132 is much larger than the field WAS value more specific to alpha-pinene. In addition, one of the FLAME-4 PTR-TOF-MS calibrations was just revised based on the work described in Hatch et al., (2016). Thus, we have revisited this in detail. We have revised the text and the species impacted by isomers are now clearly flagged in Table S3.

35

P17, L30: Old text: "Table S3 compares all 31 gases measured for Kalimantan samples in both the lab (FLAME-4) and the field. The average of the two lab EFs is within a factor of two of the field mean for 20 of 31 species, which is adequate given that a factor of two is essentially also the field coefficient of variation (n = 35). In 7 of the 11 cases with more than a factor of two difference, the lab value is actually the sum of isomers compared to a single isomer

[revised manuscript text omitted]

**R2.15:** Page 18, line 14 – you should reassure the reader that the 37% of unidentified or tentatively assigned mass 25 peaks of the NMOG mass is not going to negatively affect your assumption that you are measuring all the carbon to be factored into your EF calculations. I'm sure it's not significant, considering the major non-NMOG carbon species, but this should be recognized.

Authors: P19 L 33-35 we added: "The missing NMOG mass in the field measurements is not large enough to cause significant error in our field carbon mass balance, but would impact estimates of secondary formation of aerosol and O3 (Yokelson et al., 2013; Hatch et al., 2015)."

30

R2.16: In Table 1, Table 2, Figure 1, etc., there are numbers that are both too precise considering the standard deviations reported, and I dislike the excel-style presentation of numbers with exponents written as (e.g.) 1.67E-3. Authors: We have changed the tables so they are no longer in the excel-type format. . The number of figures in the

35 table values is to avoid round-off error, though we have changed values where the significant figures did not match: e.g. CH3I, 0.0125(0.00448) corrected to 0.0125(0.0045).

**R2.17:** Re: Table S1 – there are a handful of things that would make this table easier to digest, without having to search out other information.

Authors: We have italicized the Referees suggestions in the rest of this comment and reply point by point in un-40 italicized text.

Instead of "Y/N type" or "Y/N what" as a header, eliminate Y/N and just include the type/what or put "none" or "unknown" where applicable.

45 Au: Done.

50

Also, please spell out here what the peat fuel types are, so that I don't have to go back and find that in the paper (in *the footnote would be fine.*)

Au: They are now spelled out in each cell.

"day-mon" should be "DD-Mon". Au: Done.

Why are there some plumes included that aren't lettered? These don't seem to add anything to the paper.

55 Au: The unlettered plumes represent occasions when it was possible to quickly collect a WAS canister, but not deploy all the instruments. These samples add important emissions information.

*"seec" is not a word or a shortform (as a direction).* Au: *"seec" was changed to "sec"*

5 *For Depth of Burn, " site avg" is redundant. "site avg" is fine.* Au: We changed to "approx. site avg" and deleted the "~" where they appeared before entries.

Why is so little known about site 6?

Au: The team member collecting site data was unable to participate on the last day of the field campaign. The emissions team was able to collect some data.

*Re:* Winds – "av, max, dir" implies you'll have numeric values below. Maybe leave the "av, max, dir" part out, and just consider it a verbal description of the winds. Au: Done.

15

20

30

10

*Be consistent with spacing and vertical cell centering.* Au: This was checked.

Also, for all three supporting information tables (S1-S3), please be consistent about font size and styles and remove bold settings.

Au: The journal has requested that we use 9 pt Times New Roman with bold headings for our print tables and we assumed they want that for the supplement as well. However, Table S1 was clearly outside the normal journal format so we changed the font and font size to optimize readability.

25 *Table titles should all be uniformly sized.* Au: Please see above.

For all supplement tables, if these are being submitted as they are now in an excel file, a san serif font is likely best for readability. If you're preparing a printed document, a simple serif font is also acceptable (i.e., Times New Roman.)

Au: We have been asked to use Times New Roman 9 pt for tables by the journal so we did that. On Table S1 we used a more readable font since it is clearly a non-standard table.

**Referee #2 (Continued) Technical Comments**

R2.18: Page 5, lines 4-5 – don't us semicolons in place of commas.
 Authors: We separated the items in the list with semi colons because one item has multiple terms separated by commas: "...; charred logs, char, and ash from previous burns;"

**R2.19:** Page 5, line 26 – remove the hyphen from "at six-different peatland: : :" **Authors:** removed

**R2.20:** Page 6, line 10 – no need for "(#2)" after "This site: : :" **Authors:** deleted

45 **R2:21:** Page 11, line 12 – there is a missing or extra parenthesis here. **Authors:** The first "(" was deleted and now reads " $1/(1+\Delta CO/\Delta CO_2)$ "

**R2.22:** Page 18, line  $28 - 0.35 \pm 1 \times 10$ :  $0.10 \pm 0.10$ . Authors: fixed

50

40

**R2.23:** Page 19, line 1 – "Six of the nine: : :" **Authors:** changed to "six of the eleven"

R2.24: Page 19, line 11 – the Putra et al. paper in preparation needs to be included in the reference list.
 Authors: added

**R2.25:** Tables 1, S2 and S3: "ethyne", "ethene", "propene" (and in the text and Figure 1, where applicable). **Authors:** We think the common name for these species and many others (e.g. formaldehyde, formic acid, methanol and many more) is more or reasonably common, but added the requested formal names in the parentheses with the formula at the first appearance of these species in the text (P7).

**Response to Referee #3**

We thank the Referee for their encouraging assessment and constructive suggestions, which will improve the paper. The Referee comments are reproduced below followed by our detailed response.

**10 Anonymous Referee #3**

5

This paper presents results of a field campaign measuring emissions from burning Indonesian peat in-situ, during the intense burning during the intense 2015 El Nino event. These very challenging measurements were collected with mobile sampling set up incorporating FTIR to measure a range of gas phase species, several photoacoustic extinctiometers to measure aerosol light absorption and scattering at two wavelengths and filter and canister

15 samplers to collected integrated samples of condensed- and gas-phase species, respectively. 35 separate plumes were measured with different combinations of instruments, resulting in measurements constraining both the central tendency and the (relatively large) variability in emission factors resulting from combustion of this fuel.

As evidenced by the startlingly high PM concentrations (>3000 ug/m3) observed in nearby cities, peat combustion can and does make an enormous contribution to loadings of atmospheric aerosols and a wide range of gas-phases

- 20 species. Considering that the only extant emission factors come from a handful of laboratory burns, which a) do not capture the potential variability in emissions and, ,b) may not recreate combustion conditions observed when the fuel is in place, there is great value in the results of this study. The measurements of gas and aerosol species and aerosol optical properties appear to be carefully conducted and are well documented in this manuscript, and I particularly applaud attention to uncertainty in measurements and the resulting propagated uncertainty in EFs
- 25 (though echo the other referees' comments concerning their presentation in tables). A rather extensive effort is made to compare the results with those measured in earlier lab measurements, showing general consistency but some very significant differences. The resulting emission factors will be of great use for emission inventory development and chemical transport modeling to understand the impacts of the dramatic land use transformation taking place in this region. Therefore, the manuscript is certainly suitable for publication in ACP. Below I list several points of
- 30 clarification that would enhance the readability and utility of the manuscript.

**R3.1:** One general comment is that there is a relatively large number of portions of the text that seem overly detailed and make the paper harder to read than it might otherwise be. Ideally, these could be moved to an online supplemental section. While I understand the use of a spreadsheet for the supplement in this case, perhaps a second file could be used or some of these details moved to aid readers. Examples include: P. 6, L1-25 (description of

35 sampling sites), P. 8, L22-28 (description of PAX operating principle), P. 10, L20-28 (description of alternate data reduction approaches).

40

**Authors:** There are two schools of thought on whether to include details in the main paper, which lengthens it, or in a supplement where they may be ignored. The Referees have done an excellent job of reading the entire paper in detail, and while we agree with the Referee, the suggested supplement content would be a bit fragmented and have the unwieldy task of containing both text and spreadsheets. Thus, we prefer to skip this suggestion.

Minor Points:

**R3.2:** P9, L23-24 – This description is not clear and imprecise; for example, what is the standard deviation of smoke PM? I think I understand this to be the standard deviation of PM mass concentration, but since this is from a single filter, how is a standard deviation determined? And why is 10% of PM concentration used?

Authors: The standard deviation corresponds to the standard deviation of three replicate measurements of filter preand post-weights. The inclusion of 10% of the PM mass is a conservative estimate of the analytical uncertainty in this measurement. The reasoning is, the standard deviations of the filter weights and background PM are very small (ug). For filters with high mass loadings, the absolute error estimated on standard deviations alone would be very small (<0.1%), so we add in this relative error.

To clarify, we changed this text to read: "Uncertainty in the excess mass in the smoke plumes was propagated using the standard deviation of triplicate measurements of pre- and post-sampling filter weights, the standard deviation of background PM masses, and 10 % of the PM mass concentration, which is a conservative estimate of the analytical error associated with this measurement."

R3.3: P12, L3-4 – this is imprecise, BC does not absorb light 'proportional to frequency'.

Authors: To a first approximation sp2 hybridized carbon (including BC) absorbs light proportional to frequency
 (e.g. https://en.wikipedia.org/wiki/Aethalometer), which is mathematically equivalent to the more common statement that BC has an AAE of one (Bond and Bergstrom, 2006).

**R3.4:** P12, L25 – Unclear. What is meant by 'less close', and what do you mean 'not well-designed for comparison'?

Authors: This text has been deleted as explained in the response to Referee #2.

20 **R3.5:** P13, L11-12- While I get what you're saying, this is also imprecise. It doesn't necessarily follow that fires with both smoldering and flaming will have a linkage between MCE and EFs. I think what you mean is that in cases where you have a wider range of MCEs you tend to see a (anti) correlation between MCE and EFs, but the small range of low MCEs observed here means you don't see such a trend.

Authors: Good point. The MCE range is small, but more importantly, it is shifted below the range where it would
 help indicate the mix of flaming and smoldering, which have different products. We have changed the text as follows.

Old text: "For most biomass fires there is both flaming and smoldering and so EFs correlate with MCE, but these fires burned by smoldering only with no high MCE values (e.g. >0.9) and little or no correlation of EFs with MCE."

New text: "For most biomass fires there is both flaming and smoldering combustion and EFs for flaming compounds
 are observed to correlate with MCE while EFs for smoldering compounds (most NMOGs) tend to be anti-correlated with MCE (Burling et al., 2011). However, these fires burned by smoldering only with no high MCE values (e.g. >0.9) and little or no dependence of EFs on MCE was observed."

R3.6: P13, L13-14 – Awkward sentence, what would such characteristics be?

Authors: As noted in our response to the other Referees, we did not note a clear dependence of EFs on any peat
 characteristic. We also did not have very good capabilities to characterize the peat or map the peat types, so rather
 than discuss this in detail we have just deleted the text.

**R3.7:** P13, L18-19 – This evolution in the absorption in this plume seems to be also linked with CO emissions, with Bap(405nm) and CO very tightly correlated in the early stage of the burn, and much less so later on. Were stages of combustion typically seen? Is there any somewhat consistent trajectory in emissions during a burn? I assume that

because you measured a relatively large number of plumes that you captured emissions from a range of stages, but it would be interesting to learn of any consistency to give some insights into how laboratory tests can be more representative of combustion observed in large fuel beds in-situ.

Authors: Good point. Babs-405 and CO are actually tightly correlated throughout the sampling, but the ratio of 5 Babs-405 to CO decreases towards the end, which is consistent with an increase in the glowing to pyrolysis ratio. We're not sure what the most representative mix of these processes is, but hope we captured it with our sample size. To acknowledge this signature of the types of smoldering we added text at P13, L21: "Babs at 405 and CO remain correlated, but the ratio of Babs at 405 to CO decreases towards the end of the 5 Nov data, which is consistent with an increase in the glowing/pyrolysis ratio (Yokelson et al., 1997). Variation in the mix of these smoldering processes

10 likely causes some of the variation in EFs."

30

**R3.8:** P14, L3-4 – 'obtained closer to 500 nm' is unclear, presumably this refers to illumination wavelength? Would be best to make these MSE values more directly comparable if at all possible.

Authors: We changed "Either value of the MSE is close to MSEs obtained closer to 500 nm" to "Either value of the MSE is close to MSEs obtained at illumination wavelengths in the range 532 - 550 nm."

- 15 On the comparison: We have data at 405 and 870 nm and most of the literature MSE values are at wavelengths such as 532 or 550 nm, which is much closer to 405 nm than 870 nm. We have seen (and discuss) SSA measurements for peat aerosol at 781 nm, which is close to 870 nm and, having some idea that it was reasonable, we estimated an SSA at wavelengths between our data. We have not seen MSE measurements for peat aerosol at 405, 781 or 870 nm and are not confident about how to interpolate or otherwise estimate a value closer to  $540 \pm 10$  nm based on our data.
- 20 R3.9: P15, L16-18 – This sentence is very hard to read/understand. Overestimated by what, relative to what?

Authors: There is a tendency to time field campaigns to capture peak behavior, which makes sense, but that is not the only valid time to sample. We changed: "should not be overestimated" to "is difficult to estimate"

**R3.10:** P15, L33 – Not sure what is meant by 'significant areas' or 'deep burn depth'. Please clarify.

Authors: Upon reflection, this sentence just repeated the point of the previous sentence, which was that our measured burn depths are too high for some burned areas. We've deleted the sentence. 25

**R3.11:** P15, L35 – What is distinctly wrong about 'fire products used in top-down approaches'? This sentence is unclear.

Authors: We explained and modified the text as described in response to Referee #2. Briefly, in top-down (inverse modeling) approaches, fires and burned area can be uncertain due to factors such as uncertainty in air mass factors, emission factors, smoke injection altitude, meteorology, and the evolution of species used as constraints.

The new text on P16 L10-17 now reads: "On the other hand, burned area is likely underestimated in inventories since they rely on remote sensing data that misses some of the hotspots and burned area used in bottom-up estimates, as well as some of the fire products (e.g. CO, aerosol) used in top-down approaches. The information gap is caused by high regional cloud cover; orbital gaps; rapid growth of new vegetation, which is strongly associated

35 with shallow burn depth (Cypert, 1961; Kotze, 2013); and other factors (Lu and Sokolik, 2013; Reddington et al., 2016; Reid et al., 2013). Thus, overestimating burn depth and underestimating burned area tend to cancel when coupling these terms to estimate fuel consumption."

**R3.12:** P16, L1 – Following from previous confusing sentences, not sure what is meant by 'tend to cancel'.

Authors: In addition to the above new text, please see the detailed response to Referee #2.

**R3.13:** P19, L4-7 – This is a bit of a non-sequitur as you are talking about exposure and health impacts, and then shift to an EF comparison concerning lab/field measurements which doesn't really so much apply to these very 'high level' estimates. If anything a consistency in air toxic/PM ratio could be highlighted, as this is what you're using to estimate air toxic exposure concentrations.

5 **Authors:** Six of the air toxics were only measured in the lab study so we needed to show the relevance of the lab data if we intended to use it, especially for the two more potent pollutants measured in both settings. The Referee points out correctly that some sort of transition was needed and we have added this at P19, L4: "Two of these key species were measured in both the field and the lab and we compare the results."

The companion paper on PM will be based on a much larger, and thus more representative, set of filter data and provide a better comparison to CO and other air toxics.

**Field measurements of trace gases and aerosols emitted by peat fires in Central Kalimantan, Indonesia during the 2015 El Niño**

Chelsea E. Stockwell1,7, Thilina Jayarathne2, Mark A. Cochrane3, Kevin C. Ryan4,8, Erianto I. Putra3,5, Bambang H. Saharjo5, Ati D. Nurhayati5, Israr Albar5, Donald R. Blake6, Isobel J. Simpson6, Elizabeth A. Stone2, Robert J. Yokelson1

1University of Montana, Department of Chemistry, Missoula, 59812, USA

2University of Iowa, Department of Chemistry, Iowa City, 52242, USA

3South Dakota State University, Geospatial Sciences Center of Excellence, Brookings, 57006, USA

4United States Forest Service, Missoula Fire Sciences Laboratory (retired), Missoula, 59808, USA

5Bogor Agricultural University, Faculty of Forestry, Bogor 16680, ID

6University of California, Irvine, Department of Chemistry, Irvine, 92697, USA

7Now at: Chemical Sciences Division, NOAA Earth System Research Laboratory, Boulder, 80305, USA

8Now at: FireTree Wildland Fire Sciences, L.L.C., Missoula, 59801, USA

15 Correspondence to: R. J. Yokelson (bob.yokelson@umontana.edu)

**Abstract**

20

5

Peat fires in Southeast Asia have become a major annual source of trace gases and particles to the regional-global atmosphere. The assessment of their influence on atmospheric chemistry, climate, air quality, and health has been uncertain partly due to a lack of field measurements of the smoke characteristics. During the strong 2015 El Niño event we deployed a mobile smoke sampling team in the Indonesian province of Central Kalimantan on the island of Borneo and made the first, or rare, field measurements of trace gases, aerosol optical properties, and aerosol mass

- emissions for authentic peat fires burning at various depths in different peat types. This paper reports the trace gas and aerosol measurements obtained by Fourier transform infrared spectroscopy, whole air sampling, photoacoustic extinctiometers (405 and 870 nm), and a small subset of the data from analyses of particulate filters. The trace gas
- 25 measurements provide emission factors (EFs, g compound per kg biomass burned) for CO2, CO, CH4, non-methane hydrocarbons up to C10, 15 oxygenated organic compounds, NH3, HCN, NOx, OCS, HCl, etc.; up to ~90 gases in all. The modified combustion efficiency (MCE) of the smoke sources ranged from 0.693 to 0.835 with an average of 0.772 ± 0.053 (*n*=35) indicating essentially pure smoldering combustion and the emissions were not initially strongly lofted. The major trace gas emissions by mass (EF as g/kg) were: carbon dioxide (1564 ± 77), carbon monoxide (291 ± 49), methane (9.51 ± 4.74), hydrogen cyanide (5.75 ± 1.60), acetic acid (3.89 ± 1.65), ammonia
- $(2.86 \pm 1.00)$ , methanol  $(2.14 \pm 1.22)$ , ethane  $(1.52 \pm 0.66)$ , dihydrogen  $(1.22 \pm 1.01)$ , propylene  $(1.07 \pm 0.53)$ ,

propane (0.989  $\pm$  0.644), ethylene (0.961  $\pm$  0.528), benzene (0.954  $\pm$  0.394), formaldehyde (0.867  $\pm$  0.479), hydroxyacetone (0.860  $\pm$  0.433), furan (0.772  $\pm$  0.035), acetaldehyde (0.697  $\pm$  0.460), and acetone (0.691  $\pm$  0.356). 
[revised manuscript text omitted]

| Compound (formula)                              | Study avg (stdev) 35 plumes                                                    |
|-------------------------------------------------|--------------------------------------------------------------------------------|
| MCE                                             | 0.772(0.035)                                                                   |
| Carbon Dioxide (CO 2 )               | 1564(77)                                                                       |
| Carbon Monoxide (CO)                            | 291(49)                                                                        |
| Methane (CH 4 )                      | 9.51(4.74)                                                                     |
| Dihydrogen $(H_2)$                              | 1.22(1.01)                                                                     |
| Acetylene $(C_2H_2)$                            | 0.121(0.066)                                                                   |
| Ethylene $(C_2H_4)$                             | 0.961(0.528)                                                                   |
| Propylene $(C_3H_6)$                            | 1.07(0.53)                                                                     |
| Formaldehyde (HCHO)                             | 0.867(0.479)                                                                   |
| Methanol (CH 3 OH)                   | 2.14(1.22)                                                                     |
| Formic Acid (HCOOH)                             | 0.180(0.085)                                                                   |
| Acetic Acid (CH 3 COOH)              | 3.89(1.65)                                                                     |
| Glycolaldehyde ( $C_2H_4O_2$ )                  | 0.108(0.089)                                                                   |
| Furan ( $C_4H_4O$ )                             | 0.736(0.392)                                                                   |
| Hydroxyacetone ( $C_2H_2O_2$ )                  | 0.860(0.433)                                                                   |
| Phenol ( $C_{\epsilon}H_{\epsilon}OH$ )         | 0 419(0 226)                                                                   |
| 1 3-Butadiene ( $C_4H_2$ )                      | 0.189(0.157)                                                                   |
| Isoprene $(C_{+}H_{0})$                         | 0.0528(0.0433)5.28E.2(4.33E.2)                                                 |
| Ammonia (NH 2 )                      | 2 86(1 00)                                                                     |
| Hydrogen Cyanide (HCN)                          | 5 75(1.60)                                                                     |
| Nitrous Acid (HONO)                             | 0.208(0.059)                                                                   |
| Hydrogen chloride (HCl)                         | 0.208(0.039)                                                                   |
| Nitric Oxide (NO)                               | $\frac{0.0340(0.0203)}{0.0340(0.0203)} = \frac{0.0340(0.0203)}{0.0307(0.360)}$ |
| Carbonyl sulfide (OCS)                          | 0.507(0.500)                                                                   |
| $DMS(C \parallel S)$                            | 0.00282(0.00224)2.82E.2(2.24E.2)                                               |
| Chloromethane (CH-Cl)                           | $\frac{0.00282(0.00234)}{2.82E^{-3}(2.34E^{-3})}$                              |
| Promomethane (CH Pr)                            | 0.147(0.057)                                                                   |
| Mother indida (CILI)                            | $\frac{0.0101(0.0035)}{1.012}$                                                 |
| Dibromomothana (CH $\mathbf{Pr}$ )              | $\frac{0.0125(0.0045)}{1.25E} \frac{1.25E}{2(4.48E-5)}$                        |
| Ethena $(C, H)$                                 | $\frac{0.000104(0.000077)}{1.04E} + \frac{4(7.70E}{5})}{1.52}$                 |
| Dropana $(C, H)$                                | 1.52(0.00)                                                                     |
| i Putana (C H )                                 | 0.989(0.044)                                                                   |
| i -Butane ( $C_4 \Pi_{10}$ )             | $\frac{0.091(0.102)}{0.221(0.225)}$                                            |
| 1 Dutane (C II )                                | 0.321(0.225)                                                                   |
| i Butana (C II )                                | 0.182(0.085)                                                                   |
| t -Butelle ( $C_4 \Pi_8$ )               | 0.311(0.160)                                                                   |
| trans-2-Butene (C 4 H 8 ) | $\frac{0.0775(0.0380)}{7.75E} \frac{2(3.80E-2)}{2(3.80E-2)}$                   |
| $CIS-2$ -Butelle (C 4 $\Pi_8$ )      | 0.0615(0.0334) <del>0.13E 2(3.34E 2)</del>                              |
| i -Pentane $(C_5H_{12})$                 | 0.123(0.135)                                                                   |
| n -Pentane ( $C_5H_{12}$ )               | 0.243(0.131)                                                                   |
| $1,2$ -Propadiene ( $C_3H_4$ )                  | $\frac{0.00184(0.00227)}{1.84\pm 3(2.27\pm 3)}$                                |
| Propyne $(C_3H_4)$                              | 0.00565(0.00857) <del>5.65E 3(8.57E 3)</del>                            |
| 1-Butyne ( $C_4H_6$ )                           | 0.00198(0.00137) <del>1.98E 3(1.37E 3)</del>                            |
| 2-Butyne ( $C_4H_6$ )                           | 0.00115(0.00151) <del>1.15E 3(1.51E 3)</del>                            |
| 1,3-Butadyne ( $C_4H_2$ )                       | 0.000299(0.000242) <del>2.99E 4(2.42E 4)</del>                          |
| 1,2-Butadiene ( $C_4H_6$ )                      | 0.000615(0.000639) <del>6.15E-4(6.39E-4)</del>                          |
| 1-Pentene ( $C_5H_{10}$ )                       | 0.110(0.066)                                                                   |
| trans -2-Pentene ( $C_5H_{10}$ )         | 0.0397(0.0276) <del>3.97E 2(2.76E 2)</del>                              |
| cis-2-Pentene (C 5 H 10 ) | 0.0224(0.0152) <del>2.24E 2(1.52E 2)</del>                              |
| 3-Methyl-1-butene ( $C_5H_{10}$ )               | 0.0303(0.0198) <del>3.03E 2(1.98E 2)</del>                              |
| 2-Methyl-1-butene ( $C_5H_{10}$ )               | 0.0299(0.0161) 2.99E 2(1.61E 2)                                         |
| 2-Methyl-2-butene ( $C_5H_{10}$ )               | 0.0647(0.0372) <del>6.47E 2(3.72E 2)</del>                              |
| 2-Methyl-1-Pentene ( $C_6H_{12}$ )              | 0.109(0.076)                                                                   |
| 1,3-Pentadiene ( $C_5H_8$ )                     | 0.0198(0.0104) <del>1.98E 2(1.04E 2)</del>                              |
| 1,3-Cyclopentadiene ( $C_5H_6$ )                | 0.00998(0.00585)9.98E 3(5.85E 3)                                               |
| Cyclopentene ( $C_5H_8$ )                       | 0.0246(0.0157) <del>2.46E 2(1.57E 2)</del>                              |
| 1-Heptene ( $C_7H_{14}$ )                       | 0.0790(0.0540) 7.90E 2(5.40E 2)                                         |
|                                                 |                                                                                |

1

I

I

I

0.0652(0.0424)<del>6.52E-2(4.24E-2)</del> 1-Octene (C8H16) 1-Decene (C10H20) *n*-Hexane ( $C_6H_{14}$ ) *n*-Heptane ( $C_7H_{16}$ ) *n*-Octane ( $C_8H_{18}$ ) *n*-Nonane ( $C_9H_{20}$ ) *n*-Decane ( $C_{10}H_{22}$ ) 2,3-Dimethylbutane ( $C_6H_{14}$ ) 2-Methylpentane ( $C_6H_{14}$ ) 3-Methylpentane ( $C_6H_{14}$ ) Benzene  $(C_6H_6)$ Toluene (C7H8) Ethylbenzene (C8H10) m/p-Xylene (C8H10) o-Xylene (C8H10) Styrene ( $C_8H_8$ ) *i*-Propylbenzene ( $C_9H_{12}$ ) *n*-Propylbenzene (C9H12) 3-Ethyltoluene ( $C_9H_{12}$ ) 4-Ethyltoluene (C9H12) 2-Ethyltoluene (C9H12) 1,3,5-Trimethylbenzene (C9H12) 1,2,4-Trimethylbenzene (C9H12) 1,2,3-Trimethylbenzene (C9H12) alpha-Pinene (C10H16) *beta*-Pinene ( $C_{10}H_{16}$ ) 2-Methylfuran (C5H6O) Nitromethane (CH3NO2) Acetaldehyde (C2H4O) Butanal ( $C_4H_8O$ ) Furfural  $(C_5H_4O_2)$ Acetone ( $C_3H_6O$ ) Butanone ( $C_4H_8O$ ) Methyl vinyl ketone ( $C_4H_6O$ )

0.0498(0.0388)4.98E 2(3.88E 2) 0.143(0.087) 0.112(0.074) 0.0980(0.0690)<del>9.80E 2(6.90E 2)</del> 0.0895(0.0633)8.95E-2(6.33E-2) 0.0744(0.0509)7.44E 2(5.09E 2) 0.00531(0.00415)<del>5.31E 3(4.15E 3)</del> 0.0397(0.0358)3.97E 2(3.58E 2) 0.00931(0.00800)9.31E 3(8.00E 3) 0.954(0.394) 0.370(0.306) 0.0417(0.0202)4.17E 2(2.02E 2) 0.122(0.055) 0.103(0.059) 0.0271(0.0131)2.71E 2(1.31E 2) 0.00534(0.00374)<del>5.34E 3(3.74E 3)</del> 0.0118(0.0082)<del>1.18E 2(8.20E 3)</del> 0.0270(0.0228)2.70E 2(2.28E 2) 0.0235(0.0213)2.35E 2(2.13E 2) 0.0416(0.0335)4.16E-2(3.35E-2) 0.0108(0.0085)<del>1.08E 2(8.55E 3)</del> 0.0696(0.0552)6.96E 2(5.52E 2) 0.0639(0.0457)6.39E 2(4.57E 2) 0.00299(0.00288)2.99E 3(2.88E 3) 0.00167(0.00176)1.67E 3(1.76E 3) 0.121(0.123) 0.0601(0.0310)<del>6.01E 2(3.10E 2)</del> 0.697(0.460) 0.0238(0.0191)2.38E 2(1.91E 2) 0.124(0.116) 0.691(0.356) 0.136(0.068) 0.0569(0.0427)5.69E 2(4.27E 2)

| Plume ID>                                                           | Q                                            | R a                               | S                                            | T a                               | V                                 | $\mathbf{W}^{\mathbf{a}}$                     | $\mathbf{W}^{\mathbf{a}}$                    |                                                      |
|---------------------------------------------------------------------|----------------------------------------------|----------------------------------------------|----------------------------------------------|----------------------------------------------|-----------------------------------|-----------------------------------------------|----------------------------------------------|------------------------------------------------------|
| Date>                                                               | 5-Nov                                        | 5-Nov                                        | 5-Nov                                        | 5-Nov                                        | 6-Nov                             | 6-Nov                                         | 6-Nov                                        | PAX (7) avg
(stdev)                               |
| Filter #                                                            | 21                                           | 22                                           | 23                                           | 24                                           | 25                                | 27                                            | 28                                           |                                                      |
| EF BC (g/kg)                                                        | <del>5.23E</del>
30.00523
2.48E | <del>5.49E</del>
30.00549
2.60E | <del>5.27E</del>
30.00527
2.50E | <del>6.62E</del>
30.00662
3.14E | 8.32E
30.00832
3.95E        | 4 <del>.45E</del>
30.00445
2.11E | <del>3.22E</del>
30.00322
1.53E | 5.52E 3(1.62E
3)0.00552(0.00162)
2.61E 2(7.66E |
| EF B abs 870 (m 2 /kg)                        | $\frac{2.461}{20.0248}$                      | <del>2.001</del>
2 0.0260          | $\frac{2.501}{20.0250}$                      | $\frac{3.141}{20.0314}$                      | $\frac{3.9512}{20.0395}$          | $\frac{2.112}{20.0211}$                       | $\frac{1.05E}{20.0153}$                      | <del>3)</del> 0.0261(0.0077)                         |
| $EF B_{scat} 870 (m^2/kg)$                                          | 7.84                                         | 26.9                                         | 19.3                                         | 21.2                                         | 21.4                              | 17.9                                          | 13.5                                         | 18.3(6.1)                                            |
| EF B abs 405 (m 2 /kg)                        | 2.91                                         | 1.33                                         | 0.787                                        | 1.61                                         | 1.78                              | 0.651                                         | 0.405                                        | 1.35(0.85)                                           |
| $EF B_{scat} 405 (m^2/kg)$                                          | 46.2                                         | 60.9                                         | 37.3                                         | 78.6                                         | 52.7                              | 43.6                                          | 34.9                                         | 50.6(15.2)                                           |
| EF $B_{abs}$ 405 just BrC (m 2 /kg)                      | 2.85                                         | 1.29
4.22E                                | 0.733
5.36E                               | 1.54
6.74E                                | 1.69
8.48E              | 0.606
4-54E                                | 0.374
3.13E                               | 1.30(0.85)                                           |
| EF B abs 405 just BC (m 2 /kg)                | $\frac{20.0532}{2}$                          | <del>2</del> 0.0422                          | $\frac{20.0536}{2}$                          | $\frac{20.0674}{2}$                          | $\frac{20.0848}{2}$               | <del>2</del> 0.0454                           | $\frac{20.0313}{2}$                          | <del>2)</del> 0.0540(0.0176)                  |
| SSA 870 nm                                                          | 0.997                                        | 0.999                                        | 0.999                                        | 0.999                                        | 0.998                             | 0.999                                         | 0.999                                        | 0.998(0.001)                                         |
| SSA 405 nm                                                          | 0.941                                        | 0.979                                        | 0.979                                        | 0.980                                        | 0.967                             | 0.985                                         | 0.989                                        | 0.974(0.016)                                         |
| AAE                                                                 | 6.23                                         | 5.14                                         | 4.51                                         | 5.15                                         | 4.98                              | 4.49                                          | 4.29                                         | 4.97(0.65)                                           |
| MCE real-time                                                       | 0.726                                        | 0.763                                        | 0.773                                        | 0.778                                        | 0.824                             | 0.833                                         | 0.831                                        | 0.790(0.041)                                         |
| MCE grab sample                                                     | 0.693                                        | 0.761                                        | 0.779                                        | 0.795                                        | 0.824                             | 0.835                                         | 0.835                                        | 0.789(0.051)                                         |
| $\mathrm{EF}\mathrm{PM}_{2.5}\left(\mathrm{g/kg}\right)^\mathrm{b}$ | 19.3                                         | 21.5                                         | 17.9                                         | 29.6                                         | 24.3                              | 22.5                                          | 15.7                                         | 21.5(4.6) b                        |
| EF OC (g/kg) b                                           | 10.5                                         | 16.7                                         | 13.6                                         | 26.9                                         | 14.9                              | 17.6                                          | 11.6
8 985                                | 16.0(5.5) b                        |
| EF EC (g/kg) b                                           | 0.386                                        | 0.175                                        | 0.196                                        | 0.258                                        | 0.354                             | 0.237                                         | 020.0898                              | 0.242(0.103) b                            |
| MAC est. (405) (m 2 /g)                                  | 0.271                                        | <del>7.69⊑</del>
20.0769           | <del>3.40⊭</del>
20.0540           | <del>3./1⊭</del>
20.0571           | <del>1.14E</del>
40.114 | <del>3.45E</del>
20.0345            | <del>3.22E</del>
20.0322           | <del>9.13E 2(8.38E</del>
2) 0.0913(0.0838) |

Table 2. Aerosol emission factors and optical properties measured by the PAX and filter sampling.

a-For these plumes, PAX and filter collection times are completely in sync.

b-For these quantities an preferred average based on all the filter samples will be reported by Jayarathne et al., (2016 in prep).